# Neural Fixed-Point Acceleration for Second-order Cone Optimization Problems

## Abstract

Continuous fixed-point problems are a computational primitive in numerical computing, optimization, machine learning, and the natural and social sciences, and have recently been incorporated into deep learning models as optimization layers. Acceleration of fixed-point computations has traditionally been explored in optimization research without the use of learning. In this work, we are interested in the amortized optimization scenario, where similar optimization problems need to be solved repeatedly. We introduce neural fixed-point acceleration, a framework to automatically learn to accelerate fixed-point problems that are drawn from a distribution; a key question motivating our work is to better understand the characteristics that make neural acceleration more beneficial for some problems than others. We apply the framework to solve second-order cone programs with the Splitting Conic Solver (SCS), and evaluate on distributions of Lasso problems and Kalman filtering problems. Our main results show that we are able to get a $10\times$ performance improvement in accuracy on the Kalman filtering distribution, while those on Lasso are much more modest. We then isolate a few factors that make neural acceleration much more useful for our distributions of Kalman filtering problems than the Lasso problems. We apply a number of problem and distribution modifications on a scaled-down version of the Lasso problem distribution, adding in properties that make it structurally closer to Kalman filtering, and show when the problem distribution benefits from neural acceleration. Our experiments suggest that linear dynamical systems may be a class of optimization problems that benefit from neural acceleration.

## 1 Introduction

Given a map $f : \mathbb{R}^n \to \mathbb{R}^n$, a *fixed point* of $f$ is a point $x \in \mathbb{R}^n$ where $f(x) = x$. *Fixed-point iterations* repeatedly apply $f$ until a fixed point is reached and provably converge under assumptions of $f$ (Giles, 1987). We will refer to an optimization problem whose solution can be obtained by the convergence of fixed-point iterations as a *fixed-point problem*.

Continuous fixed-point problems are a computational primitive in numerical computing, optimization, machine learning, and the natural and social sciences. Recent work in machine learning has incorporated fixed-point computations into deep learning models as optimization layers, e.g., through differentiable convex optimization (Domke, 2012; Gould et al., 2016; Amos & Kolter, 2017; Agrawal et al., 2019; Lee et al., 2019), differentiable control (Amos et al., 2018), deep equilibrium models (Bai et al., 2019; 2020; 2022), and Sinkhorn iterations (Mena et al., 2018). Because these layers are now solving optimization problems, they become a significant computational bottleneck in training and deploying when using optimization layers. Quickly predicting solutions to these optimization problems would be a significant step in speeding up the use of optimization layers.

*Accelerating* (*i.e.* speeding up) fixed-point computations is an active area of optimization research that involves using the knowledge of prior iterates to improve the future ones. These acceleration methods improve over standard fixed-point iterations but are classically done without learning, because of the lack of theoretical

guarantees on learned solvers. However, for many real-time applications, traditional fixed-point solvers can be too slow.

On the other hand, fixed-point problems that get repeatedly solved in an application typically share a lot of structure, e.g., motion planning with noisy observations. Such an application naturally induce a distribution of fixed-point problem instances. This raises the question: can we learn to accelerate a fixed-point solver, when problem instances are drawn from a fixed distribution?

In this paper, we study the question of learning to accelerate fixed-point problem instances drawn from a distribution, which we term *neural fixed-point acceleration*; a key motivation of this work is also to better understand the characteristics that make neural acceleration more beneficial or easier for some problems than others. We are interested in the *amortized optimization* setting (Shu, 2017; Chen et al., 2021; Amos, 2022), where similar optimization problems are need to be solved repeatedly. In this setting, we aim to learn from solving historical problem instances in the distribution (available earlier), to to accelerate future problem instances (available at test time only), rather than being agnostic to this additional structure and restarting from scratch each time.

We design a framework for neural acceleration based on *learning to optimize*, *i.e.*, meta-learning (see Section 2): we learn a model that accelerates the fixed-point computations on a fixed distribution of problem instances, by repeatedly backpropagating through their unrolled computations. We build on ideas from classical acceleration: we learn a model that uses the history of past iterates to predict the next iterate.

Concretely, we first present a neural acceleration framework that learns an acceleration model as a drop-in replacement for a classical acceleration technique such as Anderson Acceleration, i.e., that takes the current and past few fixed-point iterates and predicts the next iterate at every step in the optimization. We then generalize this framework to support models that predict fixed-point iterates only at intermittent steps in the optimization. Predicting at intermittent steps only allows for greater training and inference efficiency than predicting iterates at every step, since we no longer need to backpropagate through the entire history of fixed-point iterations, but instead can use truncated gradients (Wu et al., 2018; Metz et al., 2021).

We apply this framework to the Splitting Conic Solver (O'Donoghue et al., 2016), solving two second-order cone problem distributions: one over Kalman filtering problems and another over Lasso problems. Our main results show that our best acceleration models achieve more than $10\times$ improvement in accuracy on the Kalman filtering problem instances, when compared to SCS with Anderson Acceleration; our improvement on the Lasso instances is more modest. Through the use of overparametrization (Arora et al., 2018), we are able to reduce the training time of our most expensive application, Kalman filtering, by a factor of at least 1.25.

In the second part of this paper, we isolate a few factors that make neural acceleration much better on the Kalman filtering problem instances than the Lasso problem instances. We apply a number of distribution and problem modifications to a scaled-down version of problem instances from our Lasso distribution, adding in properties that make it structurally closer to Kalman filtering. Our experiments, while only illustrative, provide evidence that the structural differences we identify (i.e., the amount and type of randomness in the problem distribution, the set of cones in the problem representation) are a few of the factors that allow the robust Kalman filtering problem to benefit more from neural acceleration. Our experiments suggest that linear dynamical systems may be a class of optimization problems that benefit from neural acceleration.

## 2 Related Work

**Acceleration methods.** There are many classic numerical methods for acceleration, such as *Anderson Acceleration* (AA) (Anderson, 1965) (also the default solver in SCS), *Broyden's method* (Broyden, 1965), and Walker & Ni (2011); Zhang et al. (2020). Most closely related to our work is that of Bai et al. (2022), who also learn an acceleration model; however, they focus on deep equilibrium models, and their model predicts the initial iterate and AA update coefficients at each step. Our approach is complementary to theirs, as we learn the full iterates at both the initial and the later steps; while this modeling makes it more challenging to accelerate similarly to the AA path, it allows us more easily to capture iterates for accelerating problems that are difficult for AA.

**Learning-to-optimize methods.** Our work is also related to the learning-to-optimize literature Shu (2017); Chen et al. (2021); Amos (2022). The meta-learning and learning-to-optimize work, *e.g.* (Li & Malik, 2016; Finn et al., 2017; Wichrowska et al., 2017; Andrychowicz et al., 2016; Metz et al., 2019; 2021; Gregor & LeCun, 2010), aim to learn better solutions to problems that arise in machine learning tasks. Bastianello et al. (2021) approximates the fixed-point iteration with the closest contractive fixed-point iteration. Ichnowski et al. (2021) use reinforcement learning to improve quadratic programming solvers. There have also been many recent works on learning-to-optimize that are application-driven, e.g., *e.g.* optimal power flow (Baker, 2020; Donti et al., 2021), combinatorial optimization (Khalil et al., 2016; Dai et al., 2017; Nair et al., 2020; Bengio et al., 2020), and solving differential equations (Li et al., 2020; Poli et al., 2020; Kochkov et al., 2021).

Most learning-to-optimize works, such as Li & Malik (2016); Andrychowicz et al. (2016); Wichrowska et al. (2017), consider only unconstrained optimization problems and it's not clear the best way of applying them to the convex and constrained conic optimization problems we consider here. We propose to connect these topics by viewing constrained optimization problems as a fixed-point procedure and learning to improve the convergence of, or accelerate, the fixed-point procedure.

In theory, all of the learning-to-optimize methods for unconstrained optimization could be applied to any fixed-point procedure by taking the perspective that we can find a fixed point by solving an unconstrained optimization problem over its residuals, *i.e.* $x^\star = \arg\min_x \|f(x) - x\|_2^2$. We propose a simple initial way of improving fixed-point computations by interleaving an MLP with the fixed-point iterations (*i.e.*, SCS updates) and hope that this perspective opens the possibility of exploring the other methods being investigated by the learning-to-optimize community, such as learning symbolic updates. One other connection to existing learning to optimize works is that many iterative methods such as gradient descent can also be seen as performing fixed-point iterations.

## 3 Background

In this section, we introduce the definitions and background necessary for the rest of the paper.

### 3.1 Preliminaries

**Fixed-point problem.** Given a map $f : \mathbb{R}^n \to \mathbb{R}^n$, a *fixed point* of $f$ is a point $x \in \mathbb{R}^n$ where $f(x) = x$. *Fixed-point iterations* repeatedly apply $f$ until a fixed point is reached and provably converge under assumptions of $f$ (Giles, 1987). We define a *fixed-point problem* to be an optimization problem that can be solved by repeatedly applying a fixed-point map $f$ until $f$ converges.

For example, consider Newton's root-finding algorithm. Let $g$ denote a function $g : \mathcal{R} \to \mathcal{R}$. The fixed-point map for finding a root of $g$ is $x_{t+1} = x_t - \frac{g(x_t)}{g'(x_t)}$, where $g'$ denotes the first derivative of $g$. An instance of this fixed-point problem might then be $x^2 = 18$.

Because we are interested in the amortized optimization scenario, where many similar problems need to be repeatedly solved, we also refer to a fixed-point problem also as a *fixed-point problem instance*.

**Convex Cone Optimization.** *Convex cone programming* is a class of optimization problems that are capable of representing *any* convex optimization problem (Nemirovski, 2007). In standard form, conic optimization involves solving the following *primal-dual* problems:

$$
\begin{aligned}
&\text{minimize } c^T x && \text{maximize } -b^T y \\
&\text{s.t. } Ax + s = b && \text{s.t. } -A^T y + r = c \\
&\quad (x, s) \in \mathbb{R}^n \times \mathcal{K} && \quad (r, y) \in \{0\}^n \times \mathcal{K}^*
\end{aligned}
\tag{1}
$$

where $x \in \mathbb{R}^n$ is the primal variable, $s \in \mathbb{R}^m$ is the primal slack variable, $y \in \mathbb{R}^m$ is the dual variable, and $r \in \mathbb{R}^n$ is the dual residual. The set $\mathcal{K} \in \mathbb{R}^m$ is a non-empty convex cone with dual cone $\mathcal{K}^* \in \mathbb{R}^m$. (We refer the reader to Boyd & Vandenberghe (2004) for an introduction to conic optimization.) In this paper, the fixed-point problems we consider are all convex cone problems.

### 3.2 The Splitting Cone Solver by O'Donoghue et al. (2016)

Our goal is to learn to accelerate a fixed-point solver for convex cone problems. The Splitting Conic Solver (SCS) (O'Donoghue et al., 2016) is a state-of-the-art fixed-point solver for convex cone problems eq. (1), so we use SCS as our (differentiable) solver in our neural acceleration framework.

In order to learn a model that can accelerate SCS, we will need to differentiate through its fixed-point mapping and use its convergence metric to define our loss. Here we describe SCS at a high level, focusing on the main points needed to understand how we differentiate through the SCS fixed-point mapping and how our metric of convergence, *the fixed-point residual*, is defined.

*Re-formulation.* In order to solve cone optimization problems with fixed-point iterations, SCS begins by converting the pair of primal-dual optimization problems into the following equivalent formulation, which finds $x, y, \tau$ to satisfy:

$$\begin{bmatrix} r \\ s \\ \kappa \end{bmatrix} = \begin{bmatrix} 0 & A^T & c \\ -A & 0 & b \\ -c^T & -b^T & 0 \end{bmatrix} \begin{bmatrix} x \\ y \\ \tau \end{bmatrix} \tag{2}$$

where $x, y, r, s$ are as in eq. (1) and $\kappa, \tau \in \mathbb{R}_+$, i.e., $\kappa, \tau$ are non-negative scalars. [1]

A solution satisfying eq. (2) can be mapped back to a solution of the cone program in eq. (1). The advantage of using this formulation is that it remains feasible even if the original pair of primal-dual problems is not feasible, and can provide certificates of primal or dual infeasibility. Grouping terms, we can also write a solution to the eq. (2) as:

$$\begin{aligned} &\text{find} \, (u, v) \\ &\text{s.t.} \; v = Qu \\ &\qquad u, v \in \mathcal{C} \times \mathcal{C}^* \end{aligned} \tag{3}$$

where $\mathcal{C} = \mathbb{R}^n \times \mathcal{K}^* \times \mathbb{R}_+$ is a cone with dual $\mathcal{C}^* = \{0\}^n \times \mathcal{K} \times \mathbb{R}_+$, and

$$u = \begin{bmatrix} x \\ y \\ \tau \end{bmatrix}, v = \begin{bmatrix} r \\ s \\ \kappa \end{bmatrix}, Q = \begin{bmatrix} 0 & A^T & c \\ -A & 0 & b \\ -c^T & -b^T & 0 \end{bmatrix}.$$

Section 2 of O'Donoghue et al. (2016) shows that any non-zero solution to eq. (2) is a solution to the original cone optimization problem in eq. (1). Thus, the optimization goal now becomes only to find a non-zero solution of eq. (2).

*Core Algorithm of SCS.* The core algorithm of SCS involves alternating two key steps: (1) projecting current iterates into an affine subspace by solving a linear system; (2) projecting the solution of the linear system onto the cone.

Let $\Pi_S(x)$ denote the Euclidean projection of $x$ to the subspace $S$. The core SCS algorithm performs iterations:

$$\tilde{u}^{k+1} = (I + Q)^{-1}(u^k + v^k) \tag{4}$$
$$u^{k+1} = \Pi_C(\tilde{u}^{k+1} - v^k) \tag{5}$$
$$v^{k+1} = v^k - \tilde{u}^{k+1} + u^{k+1} \tag{6}$$

The first step (eq. (4)) projects the current iterates into an affine subspace by solving a linear system. The second step projects the resulting iterates onto the cone $\mathcal{C}$. The third step simply updates $v^{k+1}$ with the difference $u^{k+1} - \tilde{u}^{k+1}$.

*Optimal solution.* O'Donoghue et al. (2016) prove that $u^i$ converges to its fixed point at the optimal solution, and so SCS uses $||u^{k+1} - u^k||_2$ as its *fixed-point residual*.

---

[1]Once $x, y, \tau$ are found, $r$ and $s$ are fully-specified.

### 3.3 Applications

In this work, we use Lasso and Robust Kalman filtering as our applications, and we describe below. We note that these are just two examples of second-order cone optimization problems, which also include antenna array design, filter design, portfolio optimization, robust linear progamming, and many applications in control and robotics (Alizadeh & Goldfarb, 2001; Lobo et al., 1998).

**Lasso.** The Lasso (Tibshirani, 1996) is a well-known machine learning problem formulated as follows:

$$\underset{z}{\text{minimize}} \ (1/2)||Fz - g||_2^2 + \mu||z||_1$$

where $z \in \mathbb{R}^p$, and where $F \in \mathbb{R}^{q \times p}$, $g \in \mathbb{R}^p$ and $\mu \in \mathbb{R}_+$ are data.

O'Donoghue et al. (2016) introduce the following distribution over Lasso problem instances: For each instance, (1) create a matrix $F \in \mathbb{R}^{q \times p}$ where each entry is drawn from $\mathcal{N}(0, 1)$; (2) create a vector $z^* \in \mathbb{R}^p$, also with each entry in $\mathcal{N}(0, 1)$, and set a random 90% of its entries to 0; (3) compute $g = Fz^* + w$, where $w \sim \mathcal{N}(0, 0.1)$; (4) set $\mu = 0.1||F^T g||_\infty$.

Throughout, we will refer to problem instances drawn from the above distribution as the *Lasso distribution*, or the *Lasso problem distribution*, and denote it as $\mathcal{D}_{lasso}$.

**Robust Kalman Filtering.** Our second example applies robust Kalman filtering to the problem of tracking a moving vehicle from noisy location data. Here again, we use the modeling of Diamond & Boyd (2022) as a linear dynamical system, and their method of problem generation. However, we describe the problem in slightly different notation (to avoid overlap with other mathematical notation in our work).

Let $x_t \in \mathbb{R}^n$ denote the state at time $t \in \{0 \ldots T-1\}$, and $y_t \in \mathbb{R}^r$ be the state measurement. The dynamics of the system are denoted by matrices: $F$ as the drift matrix, $G$ as the input matrix and $H$ the observation matrix. The model also allows for noise $v_t \in \mathbb{R}^r$, and input to the dynamical system $w_t \in \mathbb{R}^m$. With this, the problem model becomes:

$$\begin{aligned} \text{minimize} \quad & \Sigma_{t=0}^{N-1}(||w||_2^2 + \mu\psi_\rho(v_t)) \\ \text{s.t.} \quad & x_{t+1} = Fx_t + Gw_t, \quad t \in [0 \ldots T-1] \\ & y_t = Hx_t + v_t, \quad t \in [0 \ldots T-1] \end{aligned}$$

where our goal is to recover $x_t$ for all $t$, and where $\psi_\rho$ is the Huber function:

$$\psi_\rho(a) = \begin{cases} ||a||_2^2 & ||a||_2 \leq \rho \\ 2\rho||a||_2 - \rho^2 & ||a||_2 \geq \rho \end{cases}$$

To obtain the data for a full problem instance, the modeling simulates the system forward in time to obtain $x_t^*$ and $y_t$ for $T$ time steps. Concretely, a problem instance is generated through the following process: first, it generates $w_t^* \sim \mathcal{N}(0, 1)$, and initializes $x_0^*$ to be $\mathbf{0}$, and set $\mu$ and $\rho$ both to 2. Then, it generates noise $v_t^* \sim \mathcal{N}(0, 1)$, but increase $v_t^*$ by a factor of 20 for a randomly selected 20% time intervals. Finally, it uses $x_0^*$, $w_t^*$, $v_t^*$ together with the dynamics matrices $F$, $G$ and $H$, to simulate the system forward and obtain the observations $y_t$ for $T$ time steps. Our optimization variables for each problem instance are thus $x_t$, $w_t$ and $v_t$. For ease of reference, we include the full dynamics matrices $F$, $G$ and $H$ in Appendix A.

This problem generation process naturally induces a distribution over problem instances. Throughout, we will refer to problem instances drawn from the above distribution as the *Kalman filtering distribution*, or the *Kalman filtering problem distribution*, and denote it as $\mathcal{D}_{rkf}$.

A Lasso or a Kalman filtering problem instance can be converted to a second-order cone optimization problem in standard form (eq. (1)). In our experiments, we use CVXPY's canonicalization (Agrawal et al., 2019) to transform the problem instances (the cones corresponding to each problem type are shown in Table 1, and are discussed further in the experiments).

# 4 Neural Acceleration: A First Framework

## 4.1 Definitions

As described earlier, our focus is on the amortized optimization setting, where similar optimization problems are solved repeatedly. Let $\mathcal{D}$ denote a probability distribution of fixed-point problem instances, and a problem instance $p$ is drawn at random from $\mathcal{D}$. We define a *context* $\phi$ as the set of parameters that uniquely define a problem instance $p$. For example, in the convex cone optimization in eq. (1), $A$, $b$, $c$ and $\mathcal{K}$ uniquely define a problem, and thus, will constitute $\phi$. Throughout this work, we assume that the cones $\mathcal{K}$ are identical for all problem instances in $\mathcal{D}$, and therefore, $A$, $b$, $c$ will suffice to encode the context $\phi$. We use $f$ to denote a fixed-point map that solves every problem instance $p$ in $\mathcal{D}$; in our work, $f$ is given by the core algorithm of SCS (eq. (4) - eq. (6)). Given $f$ and a problem instance $p$ with context $\phi$ from $\mathcal{D}$, our goal is to find a fixed-point $x = f(x; \phi)$.

Since we have to solve many similar optimization problems, we aim to learn from solving historical problem instances to accelerate solving future problem instances (i.e., test instances) from the same distribution. For this reason, we measure the acceleration achieved only on the test instances, and do not explicitly take into account training time; the core assumption in amortized optimization is that the cost of training is amortized by its use on test instances, where we need to obtain fast approximate solutions.

To characterize how well a problem is solved, we use its *fixed-point residual norms*, which is defined as $\mathcal{R}(x; \phi) \stackrel{\text{def}}{=} ||x - f(x; \phi)||_2$. We use this metric as the convergence analysis of SCS (and Anderson Acceleration) are built around the fixed-point residual, as discussed in Section 3.2.

We use $\theta$ to denote the parameterization of the learning models. We define two models that will be learned:

- the *initializer* $g_\theta^{\text{init}}$ as the model that provides a starting fixed-point iterate, typically using as input the initial problem instance context $\phi$,
- the *acceleration model* $g_\theta^{\text{acc}}$ as the model that updates the fixed-point iterate at all further iterations after observing the application of the fixed-point map $f$.

## 4.2 Framework

Next, we describe our first neural fixed-point acceleration framework, shown in Alg. 1. Given a context $\phi$, we solve the fixed-point problem by maintaining the *fixed-point iterate* $x_t$ and a *hidden state* $h_t$ at each time step $t$. Then, we do the following:

- In the first time step, the initializer $g_\theta^{\text{init}}$ takes as input the context $\phi$, and provides the starting iterate and the first hidden state.
- In all further time-steps, we apply both the fixed-point map $f$ and the model $g_\theta^{\text{acc}}$ to obtain the next iterate:
  - We apply the fixed-point map $f$ on $x_t$ to obtain an intermediate iterate $\tilde{x}_{t+1} = f(x_t; \phi)$.
  - The acceleration model $g_\theta^{\text{acc}}$ uses the hidden state $h_t$, the current fixed-point iterate $x_t$, and the intermediate iterate $\tilde{x}_{t+1}$ to obtain the next fixed-point iterate $x_{t+1}$.

Note that the framework in Alg. 1 shows only the inference phase, where the learned models $g_\theta^{\text{init}}$ and $g_\theta^{\text{acc}}$ are applied to accelerate the test instances. It does not show the training phase.

We measure our acceleration by the fixed-point residuals achieved at the end of a given number of iterations (equivalent to the number of iterations required to achieve a specific residual value). Improving this metric is a necessary first step to a practical wall-clock run-time improvement, because each iteration in our framework includes only a model access in addition to the regular fixed-point iteration (see more discussion in Section 5.3).

---

**Algorithm 1** Neural fixed-point acceleration augments standard fixed-point computations with a learned initialization and updates to the iterates.

---

**Inputs:** Context $\phi$, parameters $\theta$, and fixed-point map $f$.
$[x_1, h_1] = g_\theta^{\text{init}}(\phi)$  ▷ Initial iterate and hidden state
**for** fixed-point iteration $t = 1..T$ **do**
    $\tilde{x}_{t+1} = f(x_t; \phi)$  ▷ Original fixed-point iteration
    $x_{t+1}, h_{t+1} = g_\theta^{\text{acc}}(x_t, \tilde{x}_{t+1}, h_t)$  ▷ Acceleration
**end for**

---

### 4.3 Modeling and Learning

#### 4.3.1 Modeling

**Model.** We use a standard MLP for $g_\theta^{\text{init}}$. A recurrent model is a natural choice for the acceleration model $g_\theta^{\text{acc}}$ as it encapsulates the history of iterates in the hidden state, and uses that to predict a future iterate. For additional expressivity, we also use MLPs both before and after the recurrent cell. Our precise $g_\theta^{\text{acc}}$ architecture is thus the following: first, an MLP (which we term *encoder*); second, a recurrent model such as an LSTM (Hochreiter & Schmidhuber, 1997) or GRU (Cho et al., 2014); lastly, another MLP (which we term *decoder*).

We construct the input context $\phi$ for a problem instance $p \in \mathcal{D}$ by converting $p$ into its standard form (1), and using the $A$, $b$, $c$ to define $\phi$ as follows: $\phi = [v(A); b; c]$ where $v : \mathbb{R}^{m \times n} \to \mathbb{R}^{mn}$. Note that $\phi$ is now a vector that uniquely encodes the problem $p$, since the cone $\mathcal{K}$ is fixed for all problems in $\mathcal{D}$. The parameters $\theta$ are initialized through the initialization of $g_\theta^{\text{init}}$ and $g_\theta^{\text{acc}}$.

Applying the MLP of $g_\theta^{\text{init}}$ is straightforward. For $g_\theta^{\text{acc}}$, we do the following at each iteration $t$: (1) we apply the encoder MLP on $[x_t; \tilde{x}_{t+1}]$ to produce an output $z_{e,t+1}$; (2) we apply the recurrent cell on $z_{e,t+1}$ and the hidden state $h_t$ to obtain a second output $z_{r,t+1}$ and hidden state $h_{t+1}$; (3) we then apply the decoder MLP on $z_{r,t+1}$ to obtain a third output $z_{d,t+1}$; (4) we add $z_{d,t+1}$ weighted by a factor (termed weight scaling factor) to $\tilde{x}_{t+1}$, and finally obtain $x_{t+1}$. (Note that $h_{t+1}$ is already obtained after the use of the recurrent cell.)

**Loss.** Recall that our goal is to minimize the fixed-point residual norm of a given problem instance as quickly as possible. Thus, we can set learning objective as finding the parameters that minimize the fixed-point residual norms in every iteration across the distribution $\mathcal{D}$ of fixed-point problem instances:

$$\underset{\theta}{\text{minimize}} \ \mathbb{E}_{\phi \sim \mathcal{P}(\phi)} \sum_{t < T} \mathcal{R}(x_t; \phi) / \mathcal{R}_0(\phi), \tag{7}$$

where $T$ is the maximum number of iterations to apply and $\mathcal{R}_0$ is an optional normalization factor that is useful when the fixed-point residuals have significantly different magnitudes.

We optimize eq. (7) with a gradient-based method such as Adam (Kingma & Ba, 2014). For this, we need that the fixed-point map $f(x)$ is differentiable, *i.e.* that we can compute $\nabla f(x)$. In the next section, we describe how to differentiate through the fixed-point map of SCS.

#### 4.3.2 Differentiating through SCS

Recall that the core fixed-point iteration in SCS involves alternating two key steps: (1) projecting current iterates into an affine subspace by solving a linear system; (2) projecting the iterates onto the cone. We thus need to differentiate through both these projections:

1. *Linear System Solve.* We use implicit differentiation, *e.g.* as described in Barron & Poole (2016). Further, for differentiating through SCS, for a linear system $Qu = v$, we only need to obtain the derivative $\frac{\partial u}{\partial v}$, since the fixed-point computation repeatedly solves linear systems with the same $Q$, but different $v$. This also lets us use an LU decomposition of $Q$ to speed up the computation of the original linear system as well as its derivative.

Table 1: Sizes of convex cone problems in standard form

|  | *Lasso* | *Kalman Filter* |  |  | *Lasso* | *Kalman Filter* |
|---|---|---|---|---|---|---|
| Variables $n$ | 102 | 655 |  | Zero | 0 | 350 |
| Constraints $m$ | 204 | 852 | Cones | Non-negative | 100 | 100 |
| nonzeros in $A$ | 5204 | 1652 |  | Second-order | [101, 3] | [102] + [3]×100 |

2. *Cone Projections.* We use the cone projection derivative methods developed by Ali et al. (2017); Busseti et al. (2019); we can do so because SCS also formulates the cone program as a homogeneous self-dual embedding (Ye et al., 1994).

**Normalization of the Loss.**  As described earlier, the natural choice for the learning objective is the fixed-point residual norms of SCS. However, SCS scales the iterates of feasible problems by $\tau$ for better conditioning, and this causes a serious issue when optimizing the fixed-point residuals: shrinking the iterate-scaling $\tau$ artificially decreases the fixed-point residuals, allowing $g_\theta^{\mathrm{acc}}$ to have a good loss even with poor solutions.

We eliminate this issue in SCS+Neural by normalizing each $x_t$ by its corresponding $\tau$. Thus, the fixed-point residual norm becomes the $||x_t/\tau_t - f(x_t, \phi)/\tau_{f(x_t,\phi)}||$. We are then always measuring the residual norm with $\tau = 1$ for the learning objective. (Note that this change does not modify the cone program that we are optimizing.) We show the importance of this design choice by ablating the $\tau$ normalization in Appendix D. Busseti et al. (2019) also observe this issue for the primal-dual residual map, where they propose a similar solution. In addition, with this objective, we no longer need to learn or predict from $\tau$ in the models $g_\theta^{\mathrm{init}}$ and $g_\theta^{\mathrm{acc}}$. $\tau$ is only needed when applying the fixed-point map of SCS (i.e., when computing $\tilde{x}_{t+1} = f(x_t)$), and for this, we use the $\tau$ set by SCS without any changes.

**Convergence of Neural Acceleration.**  While we do not have a theoretical analysis of convergence, we highlight that we start with SCS (an established method whose convergence is proven), and we learn parameters for an acceleration model using a loss that makes SCS converge faster for problem instances in $\mathcal{D}$.

We acknowledge that out-of-distribution problem instances may present additional challenges. However, in practice, for out-of-distribution problems, we can apply some mitigations to ensure we still provide a good solution. As an example, recall that we know that the fixed-point residual must always reduce. If our model's prediction results in increasing the fixed-point residual, we can discard the model and simply run SCS for that problem instance. Similarly, if we find that that the residuals of SCS change far more slowly than we expect at any point, we can likely detect this and restart SCS from the origin.

### 4.4 Experiments

We show experimental results on two second-order cone problems: Lasso and Robust Kalman Filtering. For ease of reference, we denote SCS accelerated with Anderson Acceleration as *SCS+AA*, and the learning-augmented SCS as *SCS+Neural*.

#### 4.4.1 Experimental Setup

**Problem distributions.**  We now describe how we instantiate the optimization problems from Section 3.3.

For Lasso, we create a training set of 100,000 problem instances and validation and test sets of 512 problems each; we draw these problem instances from the distribution described in Section 3.3, with $p = 50$, $q = 100$. As mentioned earlier, term these Lasso problem instances as the *Lasso distribution* (for ease of reference) and denote it as $\mathcal{D}_{lasso}$.

For Kalman filtering, we create a training set of 50,000 problems and validation and test sets of 500 each. We set up our dynamics matrices as in Diamond & Boyd (2022) (also included in Appendix A for completeness). We use $n = 50$ and $T = 12$ to instantiate the problem described in Section 3.3, i.e., simulating the dynamics

for 12 time-steps. Again, as mentioned earlier we term these problem instances as the *Kalman filtering distribution* and denote it as $\mathcal{D}_{rkf}$.

We obtain each problem instance in standard form (i.e., eq. (1)) through CVXPY's canonicalization (Agrawal et al., 2019). Table 1 summarizes problem sizes, types of cones, and cone sizes of these problems.

**Training and Evaluation.** We use Adam (Kingma & Ba, 2014) to train the $g_\theta^{\mathrm{init}}$ and $g_\theta^{\mathrm{acc}}$ for up to 100,000 model updates. To solve a problem instance, we perform 50 fixed-point iterations on it for both training and evaluation. We chose to perform 50 iterations because on sample problem instances from $\mathcal{D}_{lasso}$, SCS was able to achieve 2-3 orders of magnitude reduction in residuals by 50 fixed-point iterations – recall that our goal is to use learning to get a fast approximate solution, rather than a highly accurate solution.[2] For SCS+AA, we use its default history of previous 10 iterates to compute its acceleration. We perform a hyperparameter sweep with random search across the parameters of the model, Adam, and training setup, and use the best models for the results below. Table 3 in App. B shows the values used in the hyperparameter sweep, and models and hyperparameters we found for each problem.

### 4.4.2 Results

Figure 1 shows the fixed-point, primal and dual residuals for SCS, SCS+AA, and SCS+Neural. It shows the mean and standard deviation of each residual per iteration, aggregated over all test instances for each solver. We see that SCS+Neural consistently reaches a lower residual faster than SCS or SCS+AA, in the earlier iterations, but SCS+AA is able to slightly improve over SCS+Neural in the last few iterations (e.g., past iteration 40). For example, in Lasso (Figure 1a), SCS+Neural reaches a fixed-point residual of 0.001 in 25 iterations, while SCS+AA and SCS take 35 and 50 iterations and SCS respectively; moreover, improving the fixed-point residuals earlier also results in corresponding improvement in the primal/dual residuals. Our improvement for Kalman filtering (Figure 1b) is more mixed: we reach a fixed-point residual of 0.01 in 5 iterations, compared to the 30 iterations taken by SCS and SCS+AA; however, the primal/dual residuals do not show as much improvement.

In addition, SCS+AA consistently has high standard deviation, due to its well-known stability issues in early stages of the optimization (Zhang et al., 2020). However, the standard deviations of SCS (without AA) and SCS+Neural are quite similar to each other, and much lower than SCS+AA.

## 5 Neural Acceleration with Intermittent Model Access

We now describe how we generalize the framework of Section 4 to accelerate fixed-point iterations with only intermittent model access. Unrolling through the entire sequence of fixed point iterations is both computationally expensive and memory intensive, and is likely the cause of substantially slower training. We instead use truncated gradients (Wu et al., 2018; Metz et al., 2021) to update the acceleration model to improve in the local region around where it was applied. This allows us to learn by accessing the model only at intermittent iterations.

### 5.1 Framework

We now describe our acceleration framework. Since we are no longer accessing the model at each iteration, we cannot use the hidden state of the recurrent model to encode the current problem and solution state. Instead we will use the history of the iterates and the problem context as input to allow the model to effectively reconstruct the state for a particular iteration.

**Acceleration Framework.** Formally, let the *access set $A$* be the set of fixed-point iterations at which the model is accessed. The framework now needs to make the following changes: (1) apply $g_\theta^{\mathrm{acc}}$ only at iterations in the access set (we assume $g_\theta^{\mathrm{init}}$ is always applied); (2) keep the *iterate history $H_t$* with the last $k$ iterates at

---

[2]We expect that for some problems (e.g., Lasso), SCS+AA can provide a more accurate solution if the number of allowed fixed-point iterations is greatly increased, as by then, AA will have a much longer/better history to use, and therefore, better estimate an accelerated iterate.

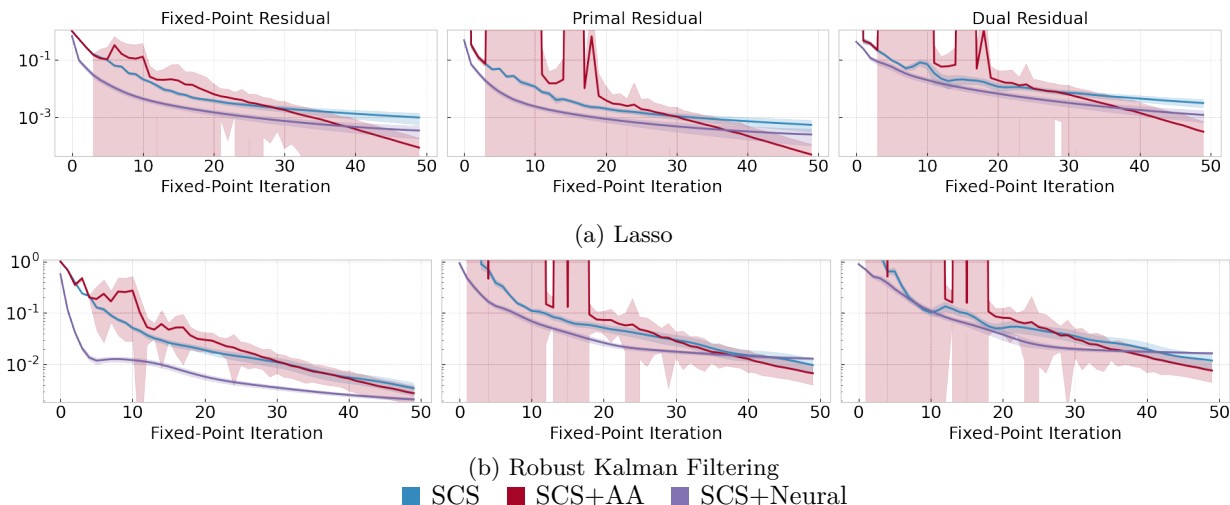

(a) Lasso

(b) Robust Kalman Filtering

■ SCS  ■ SCS+AA  ■ SCS+Neural

Figure 1: Learned acceleration models in SCS+Neural with recurrent models show modest improvements over SCS and SCS+AA.

---

**Algorithm 2** Neural fixed-point acceleration with intermittent model access

---

**Inputs:** Context $\phi$, parameters $\theta$, fixed-point map $f$, access set $A$, iterate history size $k$

$x_0 = g_\theta^{\text{init}}(\phi)$          ▷ Initial model-generated iterate

**for** fixed-point iteration $t = 1..T$ **do**

    $\tilde{x}_t = f(x_{t-1}; \phi)$          ▷ Original fixed-point iteration

    $H_t = \{x_{t-k}, \ldots x_{t-1}\}$          ▷ Update iterate history

    **if** $t \in A$ **then**

        $x_t = g_\theta^{\text{acc}}(\phi; \tilde{x}_t; H_t)$          ▷ Apply the learned acceleration model

    **else**

        $x_t = \tilde{x}_t$          ▷ Use same iterate without acceleration

    **end if**

**end for**

---

iteration $t$; (3) $g_\theta^{\text{acc}}$ now needs to use the iterate history $H_t$ and context $\phi$ in place of the hidden state to predict the next iterate.

The full acceleration framework is presented in Alg. 2. As in Section 4.2, we are given as input the problem context $\phi$, and we solve the fixed-point problem using inference with the learned models $g_\theta^{\text{init}}$ and $g_\theta^{\text{acc}}$ as well as applications of the fixed-point map $f$.

1. As before, in the first time step, the initializer $g_\theta^{\text{init}}$ takes as input the context $\phi$, and provides the starting iterate $x_0$.
2. At all further time steps $t$, we maintain the fixed-point iterate $x_t$ and the iterate history $H_t$, and do the following:

   (a) First, we apply the fixed-point map $f$ on $x_t$ to obtain an intermediate iterate $\tilde{x}_{t+1} = f(x_t; \phi)$.
   (b) Then, if $t \in A$, the acceleration model $g_\theta^{\text{acc}}$ uses the history $H_t$, the current fixed-point iterate $x_t$, and the intermediate iterate $\tilde{x}_{t+1}$ to obtain the next fixed-point iterate $x_{t+1}$; these are provided to $g_\theta^{\text{acc}}$ as a concatenation of all vectors (i.e., $[\phi; \tilde{x}_{t+1}; x_t; x_{t-1}; \ldots; x_{t-k}]$
   (c) Alternately, if $t \notin A$, we simply use the intermediate iterate unchanged as our fixed-point iterate $x_{t+1} = \tilde{x}_{t+1}$.

Once again, the framework in Alg. 2 shows only the inference phase, where the learned models are applied to accelerate the test instances. It does not show the training phase.

## 5.2 Modeling and Learning

**Model Architecture.** Throughout, we use two MLPs: one for $g_\theta^{\text{init}}$, and one for $g_\theta^{\text{acc}}$. We use an MLP for the $g_\theta^{\text{acc}}$ here, rather than a recurrent model because we found it more efficient to train MLPs rather than recurrent models, and because we only train and predict at intermittent fixed-point iterations. Our fixed-point iterates for this model are both iterates of SCS, $u$ and $v$, which we denote as $(u; v)$. We use both iterates because: (1) our access is intermittent and we do not have a hidden state, so we provide $g_\theta^{\text{acc}}$ both iterates as the iterate history; (2) predicting both $u$ and $v$ is helpful, since a mismatched $v$ can degrade what might otherwise be a good prediction of $u$.

As in Section 4, we construct the input context $\phi$ for a problem instance by converting it into its standard form (eq. (1)), and using $A$, $b$, $c$ to define $\phi$, i.e., $\phi = [v(A); b; c]$ where $v : \mathbb{R}^{m \times n} \to \mathbb{R}^{mn}$. The parameters $\theta$ are initialized through the initialization of $g_\theta^{\text{init}}$ and $g_\theta^{\text{acc}}$.

Additionally, we use *overparametrization* (Arora et al., 2018; Saunshi et al., 2020) to accelerate learning the models for the Kalman Filtering; this overparametrization increases the number of parameters in a given network without increasing the expressiveness. In our experiments, following that of Arora et al. (2018) on convolutional networks, we replace the matrix of each hidden layer of size $m \times m$ by two matrices of size $m \times m$, and the output layer of size $m \times k$ by two matrices of sizes $m \times k$ and $k \times k$ respectively.

**Loss.** As in Section 4, our loss is the *fixed-point residual norm* defined by $\mathcal{R}(x; \phi) \overset{\text{def}}{=} ||x - f(x; \phi)||_2$. With the intermittent access learning procedure, each iteration in the access set has a loss computed through fixed-point residuals. Our overall loss is the sum of all individual iteration losses; as before, we do not include the scaling factor $\tau$ in our loss (or in the models). To reduce the impact of the early iterates (whose residuals are much higher), we use the logarithm of the fixed-point norm.

We introduce one more definition before we formally define the loss. The *residual interval* $r_i$ is the number of fixed-point iterations performed after the access at iteration $i$. $r_0$ is the number of fixed-point iterations performed after the access to $g_\theta^{\text{init}}$, and we refer to all residual intervals as $R = \{r_i\}$.

Formally, we define our loss as follows. Let $y_i$ denote the point iterate at iteration $i$. When the fixed-point residuals on $y_{i+1}$ are computed for another $r_i$ steps, the loss at iteration $i$ is $\mathcal{L}_i = \sum_{j \leq r_i} \log \mathcal{R}(y_{i+j}; \phi)$. Further, for a pair of accesses $i_1, i_2 \in A$, if $r_i > i_2 - i_1$, then fixed-point residuals (and therefore the loss $\mathcal{L}_i$) include a model access for $i_2$ as well. We compute our total loss $\mathcal{L} = \sum_{i \in A} \mathcal{L}_i$ and differentiate through it.

**Learning Algorithm.** We now describe our learning algorithm, in which we train an acceleration model with intermittent access using truncated gradients. We first observe that the iterate sequence that the model needs to predict from is likely to change as the model learns. Because these are high-dimensional iterates, we use a strategy motivated by replay buffers in reinforcement learning (Sutton & Barto, 2018): we use our model to generate iterates to learn from, and then use those iterates with the fixed-point map $f$ to improve the model.

Specifically, our learning algorithm is the following:

1. We generate a sequence of iterates $H = [x_1, x_2 \ldots x_t]$ by repeatedly applying $f$, accelerating an iterate $x_i$ with our current $g_\theta^{\text{acc}}$ if $i \in A$.
2. We use this sequence of iterates to create our iterate history $H_i$ for iteration $i \in A$.
3. We use $g_\theta^{\text{acc}}$ to predict new iterates using the current iterate, iterate history, and context: $y_{i+1} = g_\theta^{\text{acc}}(x_i, H_i, \phi)$.
4. We compute fixed-point residuals on $y_{i+1}$ for another $r_i$ steps as the loss for iteration $i$: $\mathcal{L}_i = \sum_{j \leq r_i} \mathcal{R}(y_{i+j}; \phi)$. Further, for a pair of accesses $i_1, i_2 \in A$, if $r_i > i_2 - i_1$, then fixed-point residuals (and therefore the loss $\mathcal{L}_i$) include a model access for $i_2$ as well.
5. We compute our total loss $\mathcal{L} = \sum_{i \in A} \mathcal{L}_i$, differentiate with respect to $\theta$, and update $\theta$ with Adam.

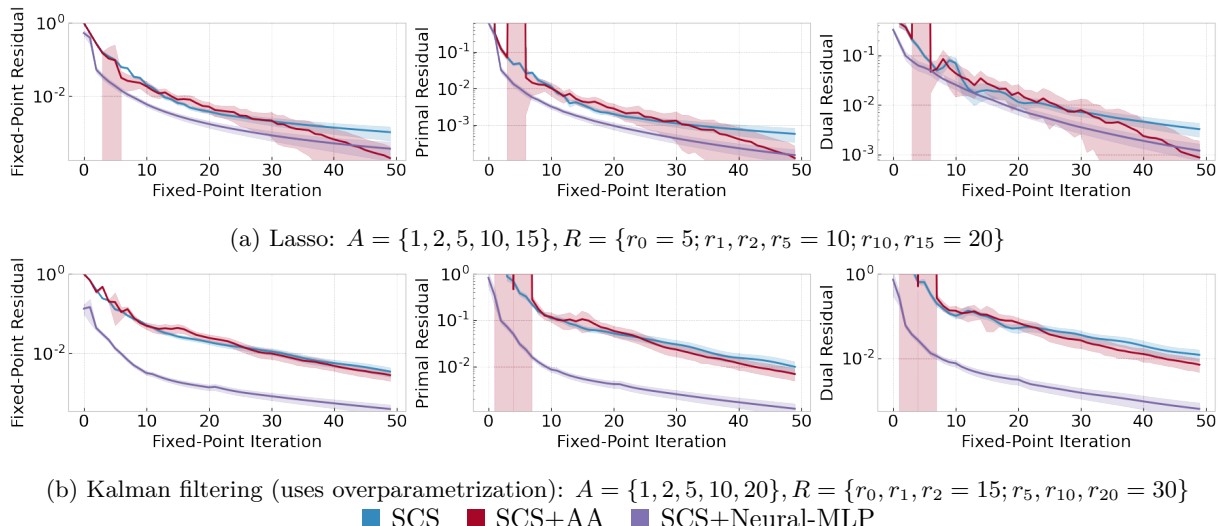

(a) Lasso: $A = \{1, 2, 5, 10, 15\}, R = \{r_0 = 5; r_1, r_2, r_5 = 10; r_{10}, r_{15} = 20\}$

(b) Kalman filtering (uses overparametrization): $A = \{1, 2, 5, 10, 20\}, R = \{r_0, r_1, r_2 = 15; r_5, r_{10}, r_{20} = 30\}$

■ SCS ■ SCS+AA ■ SCS+Neural-MLP

Figure 2: Learned acceleration models in SCS+Neural-MLP improve on upon SCS/SCS+AA to reach a better solution with only a few intermittent model accesses.

## 5.3 Experiments

We show experimental results on the same two second-order cone problems of Section 4: Lasso and Robust Kalman Filtering. To distinguish the intermittent access acceleration model from the recurrent model, we will refer to it as *SCS+Neural-MLP*.

### 5.3.1 Experimental Setup

Our experimental setup here is identical to that of Section 4 to facilitate a comparison. The primary changes come from having to additionally set up the access set and residual intervals for the model. For SCS+Neural-MLP, we allow the access set to have 5 accesses to $g_\theta^{\text{acc}}$, in addition to the access to $g_\theta^{\text{init}}$. Further, for both SCS+Neural-MLP and SCS+AA, we use a history of 5 iterates[3].

Our hyperparameter sweep now includes access set and residual intervals, which are key design elements that affect the accuracy and efficiency of the models. To efficiently find a good hyperparameter search space, we performed some ablations to identify heuristics in how access sets and and residual intervals affect model accuracy – in general, we found that earlier accesses are more useful than later ones, and it is useful for the residual intervals to be at least long enough so that they go a few iterations past the next model access; for some applications, the residual intervals need to be much longer than others. We then performed a hyperparameter sweep with random search using these heuristics as a guideline (see Table 4 in App. B for the values and heuristic dependencies used in the hyperparameter sweep). Our results use the best choices that we found for each problem. All experiments are run with 3 seeds for repeatability, and each metric is aggregated over all test set instances and all runs.

### 5.3.2 Results

Figure 2 shows the fixed-point, primal and dual residuals for each problem, similar to Figure 1. Below, we discuss our results for each problem individually.

**Lasso.** Our models for $g_\theta^{\text{init}}$ and $g_\theta^{\text{acc}}$ both have 3 hidden layers, with 2560 and 5120 units respectively, all with ReLU non-linearities. In addition, as mentioned earlier, the accuracy of the learned model strongly depends on the choice of the access sets and the residual intervals. Through extensive hyperparameter

---

[3]Note that these SCS+AA results with a history of 5 iterates are nearly identical in averages to those of Section 4 (with the default history of 10 iterates), but the reduced history greatly reduces its variance of SCS+AA.

|  | Lasso (orig.) | Lasso: $q = 40$ | Lasso: $q = 60$ | dynamic Lasso (2-step) | dynamic Lasso (3-step) |
|---|---|---|---|---|---|
| Total variables | 22 | 22 | 22 | 53 | 83 |
| Constraints | 44 | 64 | 84 | 105 | 165 |
| nnz in $A$ | 244 | 444 | 644 | 525 | 805 |
| Zero | 0 | 0 | 0 | 10 | 20 |
| Non-negative | 20 | 20 | 20 | 40 | 60 |
| Second-order | [21, 3] | [41, 3] | [61, 3] | [41, 3, 11] | [61, 3, 21] |

Table 2: Lasso variants for Section 6: problem sizes when in standard form

search, we found the following access sets and residual intervals to work well for Lasso the access set is $A = \{1, 2, 5, 10, 15\}$, with the residual interval at iteration 0 $r_0 = 5$, at iterations 1, 2, 5 $r_1, r_2, r_5$ all as 10, and at iterations 10, 15 $r_{10}, r_{15}$ as 20. For ease of notation, we denote this set of residual intervals as $R = \{r_0 = 5; r_1, r_2, r_5 = 10; r_{10}, r_{15} = 20\}$ in Figure 2. For clarity, we also summarize the complete model details in Table 6 in App. B.

Figure 2a shows the residuals of SCS+Neural-MLP; these are similar to the recurrent models in Section 4, but they require only 5 model accesses compared to the 50 required by the recurrent models. SCS+Neural-MLP is able to improve on the residuals of SCS by as many as 20 iterations (*i.e.*, by iteration 20, SCS+Neural-MLP has improved over the residuals reached by SCS at iteration 40) and over SCS+AA by as many as 12 iterations (SCS+AA reaches the same residuals around iteration 32). SCS+Neural-MLP maintains all its improvement over SCS across most of the 50 iterations, but its improvement over SCS+AA starts to degrade past iteration 35 and disappears completely around iteration 45. Unlike Kalman filtering, overparametrization does not help improve the neural acceleration model for Lasso.

**Kalman Filtering.** For Kalman filtering, we use a 3-layer network for $g_\theta^{\text{init}}$ and $g_\theta^{\text{acc}}$. We use 5120 units in the hidden layer for both $g_\theta^{\text{init}}$ and $g_\theta^{\text{acc}}$, and we use ReLU non-linearities. We also use overparametrization in the hidden layer and the output layer, as described in Section 5.2 – we add one overparametrized layer each before the hidden layer and the output layer. Our best access set is $A = \{1, 2, 5, 10, 20\}$. We use a residual interval of 15 for the accesses at iterations 0, 1 and 2, and we use 30 for accesses at 5, 10, and 20. For ease of notation, we denote these as $R = \{r_0, r_1, r_2 = 15; r_5, r_{10}, r_{20} = 30\}$. For clarity, we also summarize the complete model details in Table 6 in App. B.

Our results are shown in Figure 2b. We note we are able to improve the convergence of all three residuals for SCS+Neural-MLP over SCS/SCS+AA by almost 5× fewer iterations: by iteration 10, SCS+Neural-MLP has already reached the same residuals (on average) that SCS and SCS+AA reach by iteration 50. This is unlike the recurrent models (shown in 1b in Section 4), where only the fixed-point residual has improved by iteration 10; the corresponding improvement in primal/dual residuals is very little. Further, unlike the recurrent models, the improvement over SCS and SCS+AA remains nearly as high at iteration 50 as it is at iteration 1 – indeed, applying SCS past iteration 20 (the last access in A) continues to maintain the improvement gained from the model. Indeed, the improvement far exceeds that of recurrent models in Section 4 – the primal/dual residuals for MLP models at iteration 10 are 10× already smaller than the recurrent models; at 50 iterations, all three residuals of SCS+Neural-MLP are at least 10× smaller than the recurrent models.

Thus, we see that the increased training efficiency achieved by using truncated gradients and MLPs through the intermittent access framework have enabled significant improvements over SCS/SCS+AA on Kalman filtering. In our experiments, overparametrization turns out to be crucial for the Kalman filtering results. The overparametrized model achieved the results shown in 70k-80k training iterations, while the regular model did not obtain similar results in 100k iterations.

We discuss our implementation run-time and its caveats in Section 5.3.3.

### 5.3.3 Discussion

In this section, we discuss two aspects of our experimental results: first, the difference in improvements between Lasso and Robust Kalman filtering, and second, some caveats on our implementation run-time.

**Differences between Lasso and Kalman filtering.**  Our results show that SCS+Neural-MLP is able to accelerate the Kalman filtering problem distribution $\mathcal{D}_{rkf}$ much better than the Lasso problem distribution $\mathcal{D}_{lasso}$. We note that using the same models, overparametrization, and hyperparameters as Kalman filtering for Lasso does not result in any noticeable improvement over the results in Figure 2. A natural question this raises is why this improvement is possible for Kalman filtering, but not Lasso.

To explore this question, we begin by identifying a few structural differences between these two problems:

- *Distributions of the A, b, c.* For Kalman Filtering, $A$ is identical across all problem instances; only the $b$ varies, and only in the entries corresponding to the zero cone. For Lasso, $A, b$ and $c$ are all different across the problem instances; however, $c$ is identical subject to a normalization. In addition, $b$ varies in the entries corresponding to second-order cone of size 101.
- *Cone structure.*  The Kalman filtering problem has a zero cone of size 100 (from its equality constraints), while the Lasso problem has no zero cone. In addition, while both problems have second-order cones, the Robust Kalman Filtering has many (100) second-order cones of dimension 3, and one larger cone of dimension 102. The Lasso, instead, has two second-order cones, one of dimension 101, and one of size 3. This also induces a substantial difference in the sparsity and block-structure of the source matrices $A$ of the two problems.

These differences lead to two major reasons why neural acceleration is more beneficial for Kalman filtering. First, the Kalman filtering problem is learning to solve for the same $A$ with each of different instances; thus, this provides many more training instances and iterations for the same $A$, and reduces the amount of randomness in the problem. Second, the Kalman filtering problem has to fit a large zero cone; geometrically, the zero cone is a point, more restrictive than a second-order cone, and so all the algorithms will need to fit the solution exactly; this makes the problem harder for SCS and SCS+AA.

Based on these insights, in Section 6, we design a series of modifications that convert a Lasso problem distribution $\mathcal{D}_{lasso}$ to a linear dynamical optimization problem like Kalman filtering $\mathcal{D}_{rkf}$, and show that these modifications improve the benefit of neural acceleration.

**Caveats on our Implementation Run-time.**  Our current wall-clock run-times do not compete with SCS, since we implemented an unoptimized version in Python for easier exploratory analysis. This is because our focus in this work is only to demonstrate a proof-of-concept by learning models and identifying problem/distribution classes that may benefit from neural acceleration (see Section 6). In contrast, SCS has a highly-optimized implementation written in C.

Our experiments are run on 2.20GHz Intel Xeon CPU E5-2698 v4 with 80 cores, and with Tesla V100-SXM2-32GB GPUs (we use one GPU per experiment). Concretely, for Kalman filtering, SCS takes 1-2ms per problem instance on average, and SCS+AA takes 3ms on average for 50 iterations. For Lasso, SCS and SCS+AA take 3ms and 4ms per problem instance on average. On the other hand, our run-time for Kalman filtering is around 4 seconds per batch with 50 iterations, and for Lasso is 2.3s per batch for 50 iterations. However, a significant portion of this increase is due to the limitations of our Python implementation. For example, for Kalman filtering, our Python implementation of the SCS fixed-point updates (eq. (4) to eq. (6)) by themselves take around 1.3-1.7s (compared to the milliseconds of SCS/SCS+AA), even though these steps perform the exact same operations as SCS. We note that the additional computational operations in neural acceleration is just model inference (along with the $\tau$ normalization) at intermittent iterations, and in our Kalman filtering problems, model inference time takes only 4ms-4.5ms per batch (recall that we have 5 model accesses and an inital model access; each of these accesses takes 0.5-0.8ms typically). [4]

---

[4] Only our model is run on the GPU, with all other calculations on the CPU. SCS/SCS+AA is also run on the CPU. When the model is also run on the CPU, the total model access to solve a batch takes around 0.35-0.5s (a single model access typically takes 30ms-80ms on the CPU). Note this is still only a small fraction of the total run-time of around 4s, and all operations are still batched.

This breakdown of run-times, where the additional compute operations in neural acceleration only take a small fraction of the total run-time, suggests a path towards a faster implementation for sufficiently large problem size, e.g., one could use the optimized C implementation SCS for computing the fixed-point updates, and integrate a trained model with it. In our sample problems, the neural network evaluation per-instance takes longer than SCS or SCS+AA; however, when given much larger problem instances with relatively little randomness in their distributions, neural network evaluations that allow many fewer fixed-point iterations may be much faster than solving the problem from scratch, especially if large matrix inversions are required.

Further, note that for Kalman filtering, SCS+AA barely shows any acceleration improvement over SCS (with iterate history sizes of both 5 and 10). Recall that SCS+AA and other AA variants use linear combinations of the historical iterates to predict further iterates. Neural networks thus have a representational advantage over AA variants, because they can represent a more complex class of functions, and thus potentially accelerate problems that cannot be accelerated with AA variants.

For these reasons, we believe our work is a first step towards practical neural acceleration with wall-clock time improvement – a significant improvement in the number of fixed-point iterations is a necessary first step in this kind of framework. Our results – that SCS+Neural is able to make an improvement where AA does not – suggest classes of problems that may benefit from neural acceleration, and we discuss this further in Section 6.

Our model training took a day for the Lasso models and 3-7 days for the Kalman filtering (3 days when the most expensive operations were optimized with PyTorch JIT; 7 days without the optimizations).

## 6 Variations between Optimization Problems

In this section, we explore some of the characteristics of second-order cone problem distributions that allow for the gain of significant improvements with neural acceleration. For this analysis, we begin with the original Lasso problem distribution $\mathcal{D}_{lasso}$, and examine whether, by applying modifications to either just the distribution or the optimization problem itself, the resulting problem distribution is easier for or benefits more from learned acceleration.

To simplify our analysis, we use a much smaller problem size (than Sections 4 and 5) from the Lasso distribution. We see that even these smaller problems are not easy for neural acceleration, and even when using the same model as the larger sized Lasso problems. Each modification to $\mathcal{D}_{lasso}$ then changes the optimization problem and/or distribution such that it adds some properties of the Kalman filtering distribution $\mathcal{D}_{rkf}$, and we train new neural acceleration models on the new problem distributions. This allows us to gain insight about the characteristics that make a problem distribution better for neural acceleration. Since these experiments are on smaller-sized problems, rather than the original Lasso and Kalman filtering problem distributions, we emphasize that these results are only indicative of the underlying properties. Nevertheless, our results suggest that linear dynamical systems may be a class of optimization problems that benefits from neural acceleration.

Our experimental setup for this section follows Section 5.3, with one difference: since the optimization problem distribution varies in each experiment, we will describe those in their respective sections. However, for ease of reference, Table 2 shows the problem sizes for the different Lasso variants when they are represented in standard form.

### 6.1 Optimization Problems of the Original Distribution

Our first experiment repeats the distribution $\mathcal{D}_{lasso}$ of Section 5 on the smaller problem size, to provide a baseline for the remaining experiments. For these experiments, we use $p = 10, q = 20$, i.e. 10 variables and 20 observations. We also observe that while each problem instance is generated with an underlying solution $z^*$, the addition of the noise $w$ changes the optimization problem, and depending on the accuracy of the required residuals, $z^*$ may no longer suffice as a solution.

Figure 3 shows the results of training SCS+Neural-MLP for this smaller Lasso distribution. We see that there is only a very modest improvement over SCS, most of which is obtained around fixed-point iteration 5,

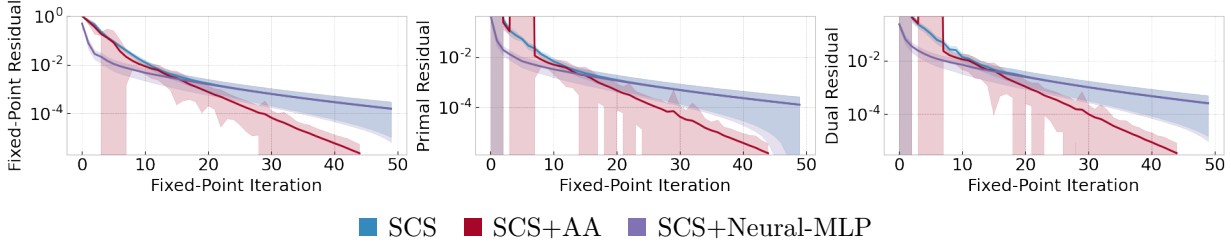

Figure 3: Original Lasso with $p = 10$, $q = 20$. Learned model uses $A = \{1, 2, 5, 10\}$, $r_i = 10$ for all $i$.

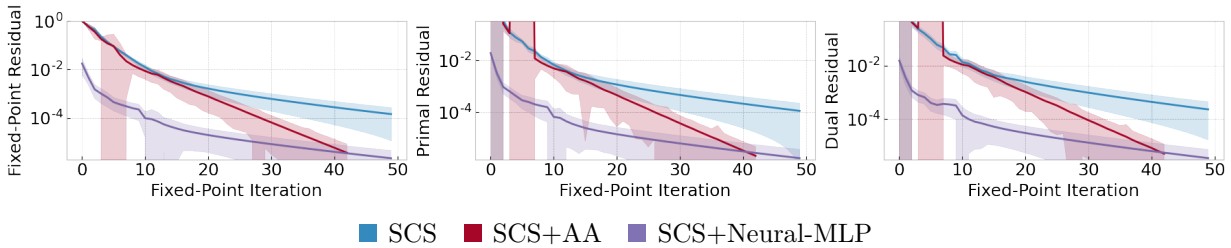

Figure 4: Reducing the randomness in the Lasso problem distribution by keeping the source matrix $F$ fixed, leading to a fixed $A$ matrix in standard form. Learned model uses $A = \{1, 2, 5, 10\}$, $r_i = 10$ for all $i$.

.

and is fully lost after fixed-point iteration 10. We observe also that the SCS+Neural-MLP can predict to a obtained to a residual of $10^{-2}$, and no improvement is obtained after that.

In Appendix C.1, we show that this trend does not change significantly for this distribution as the sparsity of the solution changes, i.e., as the number of non-zero values in $z^*$ (termed the *density* $\rho$) changes from 0.1 to 0.3 and 0.5.

## 6.2 Reducing the Randomness in the Distribution

We now examine the effect of reducing the amount of randomness in the distribution $\mathcal{D}_{lasso}$. Recall that the original Lasso distribution $\mathcal{D}_{lasso}$ has three independent components of randomly generated data: the source matrix $F$, solution $z^*$, noise $w$. (Note also that the quantity $\mu$ depends on both $F$ and $z^*$, and affects the objective of the optimization problem in standard form.)

In our experiments below, we show the effect of keeping the source matrix $F$ fixed (Section 6.2.1), and of reducing the noise in the distribution (Section 6.2.2). In App. C.2, we also include results that show that keeping the solution or the noise fixed does not affect the neural acceleration.

### 6.2.1 Fixed Source Matrix

In this experiment, we keep the source matrix $F$ fixed throughout the problem distribution: i.e., we generate one source matrix $F$, and use it to generate all problem instances, drawing $z^*$ and $w$ at random as before. To ensure that we observe general trends, rather than dependencies on any single source matrix, we aggregate our results over problem distributions for 10 different source matrices $F$.

Figure 4 shows the results for SCS, SCS-AA, and SCS+Neural-MLP on this modified distribution. We note that the performance of SCS and SCS-AA does not change noticeably compared to Figure 3, but SCS+Neural-MLP has a dramatic improvement: now, SCS+Neural-MLP improves over SCS at all 50 fixed-point iterations, typically by over an order of magnitude, and it gets primal/dual residuals as low as $10^{-5}$ in around 35-40 iterations. SCS+Neural-MLP even improves substantially over SCS-AA as well over the first 30 fixed-point iterations, and for the first 10 iterations, it maintains an improvement of 2 orders of magnitude over SCS-AA as well.

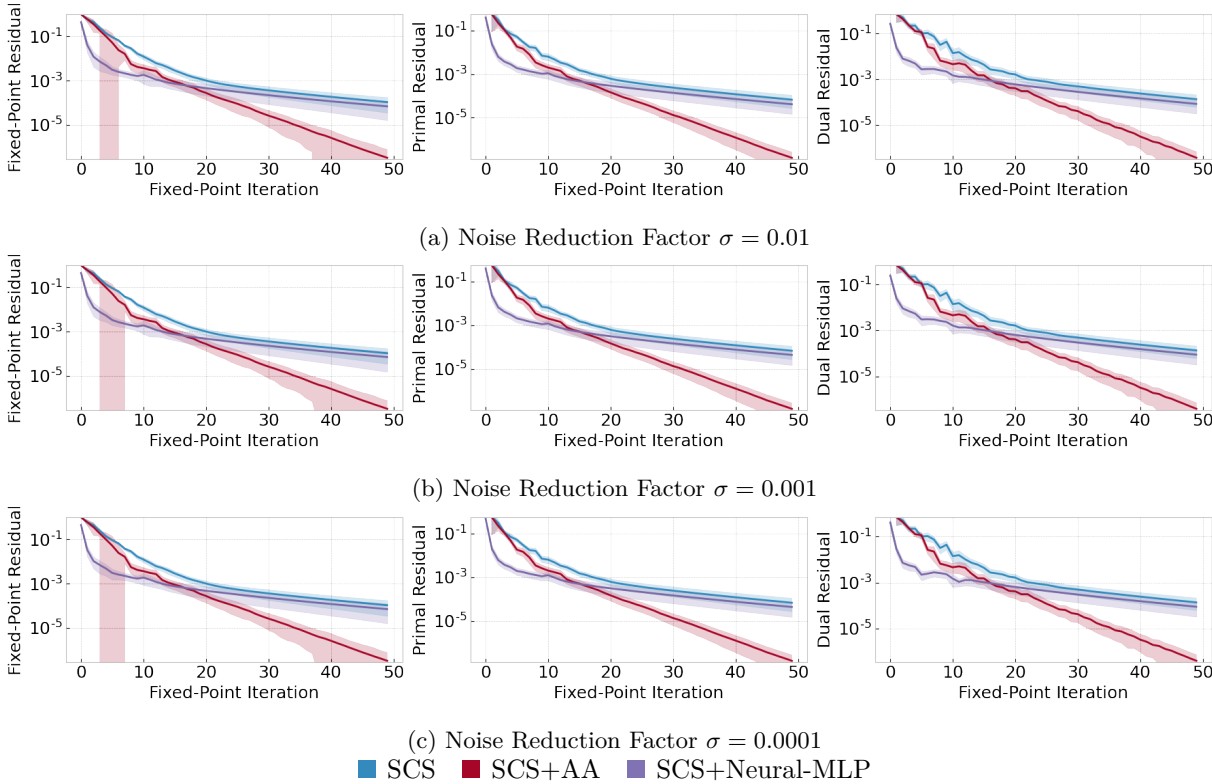

(a) Noise Reduction Factor $\sigma = 0.01$

(b) Noise Reduction Factor $\sigma = 0.001$

(c) Noise Reduction Factor $\sigma = 0.0001$

■ SCS   ■ SCS+AA   ■ SCS+Neural-MLP

Figure 5: Reduced noise in Lasso distribution. Learned models use $A = \{1, 2, 5, 10\}, r_i = 10$ for all $i$.

### 6.2.2  Reduced Noise

Our next experiment shows that just reducing the noise in the original Lasso distribution (while allowing the source matrix $F$ to change) is not sufficient to learn to optimize well.

For this experiment, we modify the problem distribution as follows: we generate $F$, $z^*$ and $w$ as earlier, but we define $g = Fz^* + \sigma w$, where $\sigma$ is a multiplicative factor that reduces the impact of the noise on the final solution. In our experiment, we use $\sigma = 10^{-2}, 10^{-3}, 10^{-4}$. Note that in our original distribution $\sigma = 0.1$.

Fig. 5 shows the results. We see that for all three $\sigma$ values, there is an improvement over the original distribution, as the SCS+Neural-MLP is now able to obtain improvements over SCS to at least $10^{-3}$, which is achieved around fixed-point iteration 10. We note also that there does not appear to be a noticeable difference between SCS+Neural-MLP's performance on the different $\sigma$ values: $\sigma = 0.01$ already obtains the maximum improvement if only the noise multiplication factor is reduced.

These results suggest that even the $F$ and $z^*$ in the Lasso distribution have too much variability for this particular model size/design to learn to optimize well.

### 6.3  From Lasso to a Linear Dynamical System

Our last set of experiments explore the ease of neural acceleration when the Lasso problem we have studied above is converted to a linear dynamical system, like the Kalman filtering problem. A linear dynamical system evolves (noisily) over time, so we will have many more observations on the same matrix. However, more observations typically reduce the space of valid good solutions (since the noise in our problem distributions are generated independently), which might make it easier for both SCS and SCS+Neural to find the optimization solution $z$.

To isolate the impact of the dynamical system from the increased number of observations, we first increase the number of observations with the variables remaining static, in Section 6.3.1. Then, in Section 6.3.2, we

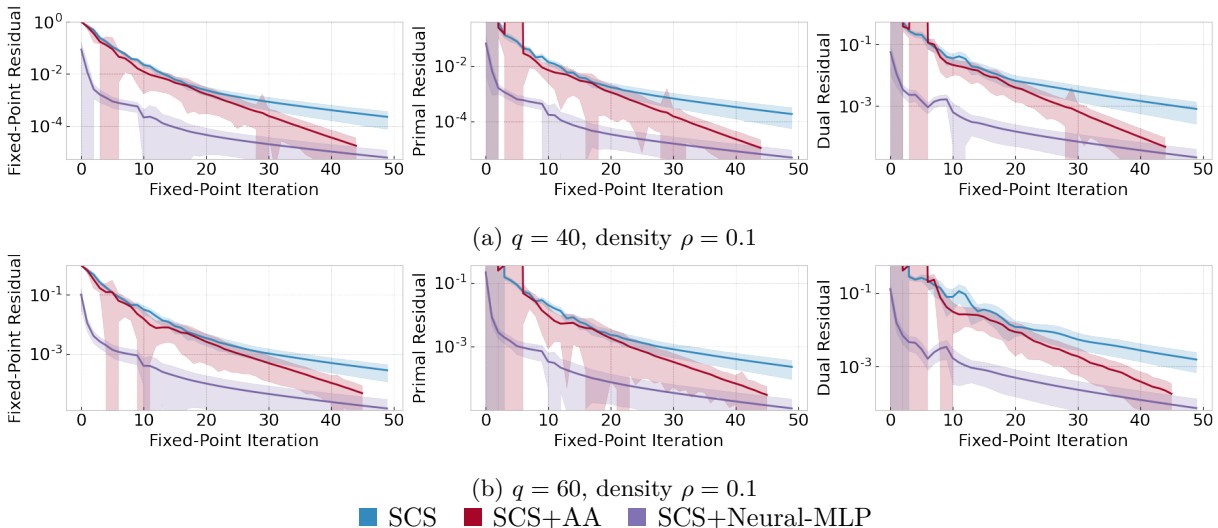

Figure 6: Increased observations for fixed source Lasso distributions. Learned models use $A = \{1, 2, 5, 10\}, r_i = 10$ for all $i$.

modify our problem into a linear dynamical system by allowing the solution to move gradually in a defined manner as well over time steps, and obtain one set of observations at each time step.

### 6.3.1 Static Solution

In this experiment, we use a Lasso problem distribution very similar to Section 6.2.1, but with increased observations. The distribution of Section 6.2.1 used fixed source matrices of size $q \times p$ where $q = 20$ and $p = 10$. Here we increase the number of observations to $q = 40$ and $q = 60$ respectively.

Figure 6 shows the results for these problem distributions. We observe that while the residuals for all three algorithms (SCS, SCS-AA and SCS+Neural-MLP) are larger than in Figure 4, the improvement of the learned models of SCS/SCS-AA is slightly increased, exceeding 2 orders of magnitude throughout. The higher residuals may likely be because having more observations $q$ makes it harder for SCS/SCS-AA to fit the noise as well in the optimization solution $z$. The results for $\rho = 0.3$ and $\rho = 0.5$ are similar, and hence we do not include them here.

### 6.3.2 Dynamic Solution

In our final experiment, we modify the Lasso into a linear dynamical system, simulating the (defined) movement of the Lasso solution over multiple time steps, and show its impact on the benefits of neural acceleration.

**Problem Generation.** Concretely, we modify our problem generation process as follows:

- We compute a source matrix $F$, and an initial solution $z_1^*$ as before, where $z_1^*$ has the required density $\rho$. We compute $w_1, w_2 \ldots$ as the noise, as independent vectors for each required step.
- For each step, we move the solution $z^*$ forward by a defined step size $\delta$, e.g., $z_{i+1}^* = z_i^* + \delta$. Only the non-zero entries of $z_i^*$ are changed in order to maintain sparsity.
- We then compute $g_i = F z_i^* + w_i$ as the output solution for each required step. As is typical in optimization problems derived from linear dynamical systems, we penalize all the noise by a reduced factor of $\sigma$.

With this process, we define our optimization problem as follows. Let $H$ denote the source matrix created by repeating $F$ for time $t$ number of times in a block structure; let $g$ denote the vector of observations $[g_1; g_2 \ldots g_t]$, and let $z$ denote the vector of all the solutions $[z_1; z_2 \ldots z_t]$; and let $v$ denote the vector of all

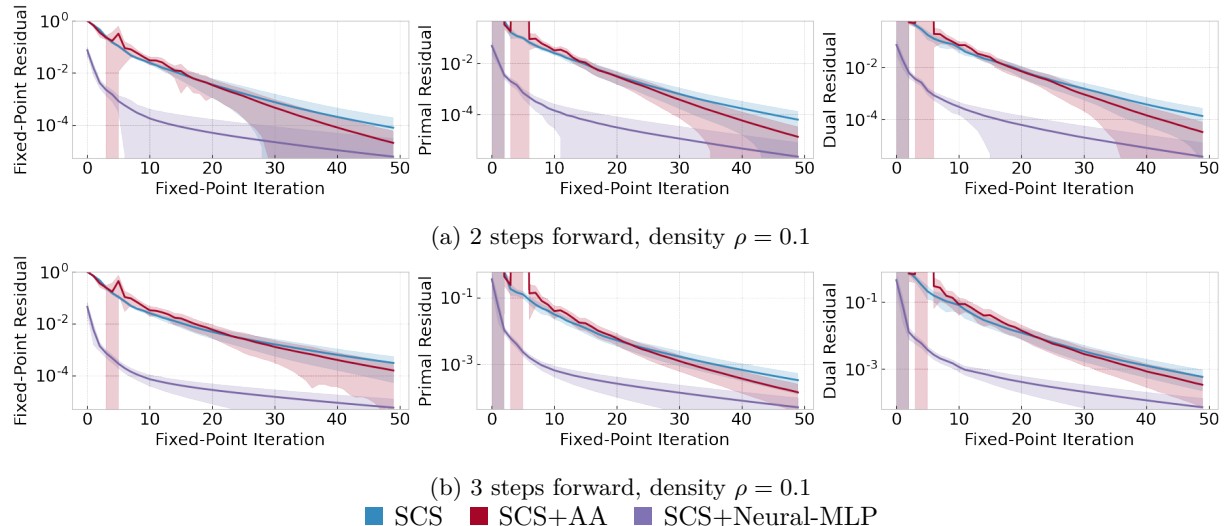

(a) 2 steps forward, density $\rho = 0.1$

(b) 3 steps forward, density $\rho = 0.1$

■ SCS  ■ SCS+AA  ■ SCS+Neural-MLP

Figure 7: A simple linear dynamical system that simulates the Lasso solution moving in time. The problem distribution has a fixed source matrix $F$, but varying $z^*$ per instance, and varying $w$ per instance and per time step. Learned models use $A = \{1, 2, 5, 10\}, r_i = 10$ for all $i$.

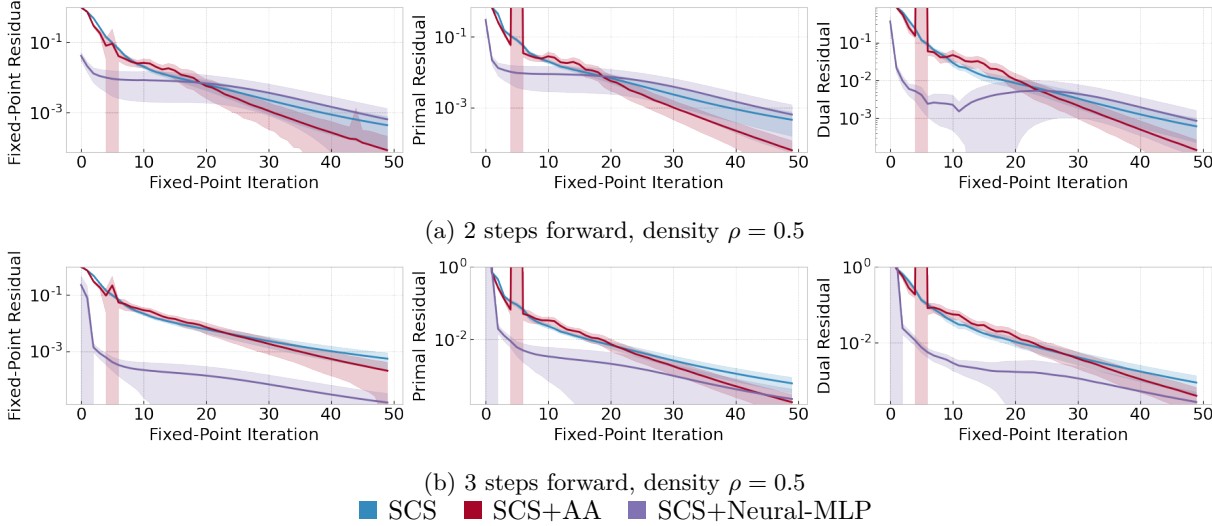

(a) 2 steps forward, density $\rho = 0.5$

(b) 3 steps forward, density $\rho = 0.5$

■ SCS  ■ SCS+AA  ■ SCS+Neural-MLP

Figure 8: Linear dynamical system simulating the Lasso forward in time with density $\rho = 0.5$. Learned models use $A = \{1, 2, 5, 10\}, r_i = 10$ for all $i$.

noise $[v_1; v_2 \ldots v_n]$.

$$\text{minimize} \, ||Hz - g||_2^2 + \lambda ||z||_1^2 + ||v||_2$$
$$\text{s.t.} \, z_{i+1} = z_i + \delta + \beta v_i$$

In our experiments, we set $\delta = 0.1$ and $\beta = 0.005$. All other parameters remain as before from the Lasso problem earlier.

**Results.** Figure 7 shows the results on this problem distribution for $\rho = 0.1$ (the results for $\rho = 0.3$ are similar); we have simulated the linear dynamical system described above 2 steps and 3 steps forward to obtain the problem distributions. We first note that this problem distribution is, overall, harder for SCS-AA to improve over SCS – note that the residuals obtained by SCS-AA are very close to that of SCS. However,

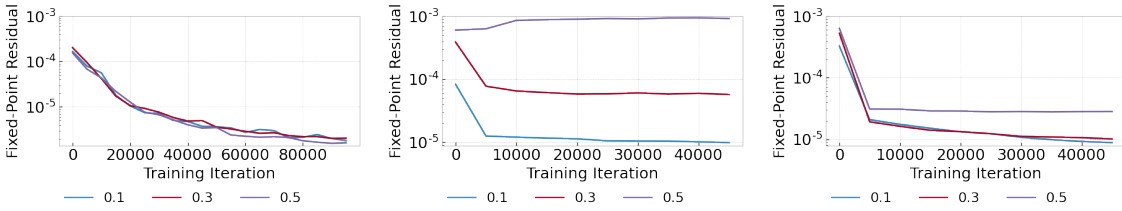

(a) Lasso: fixed source matrix $F$    (b) Dynamic Lasso: 2 steps forward    (c) Dynamic Lasso: 3 steps forward

Figure 9: Residuals as a function of training iterations. Learned models use $A = \{1, 2, 5, 10\}, r_i = 10$ for all $i$.

SCS+Neural-MLP is able to substantially improve on SCS and SCS-AA, by well over 2 orders of magnitude. Indeed, the relative improvement over SCS exceeds even that of the fixed source matrix Lasso distributions in 6.2.1.

Next, Figure 8 shows the results when $\rho = 0.5$. Note that here, the results are more mixed: when there are only 2 steps, the model learns very poorly, but with 3 steps, the model is able to learn quite well, although not as well as in Figure 7, with $\rho = 0.1$.

Finally, we explore how quickly SCS+Neural-MLP learns a good model for the different variations in Figure 9. We see that for the dynamic Lasso (with 3 steps), SCS+Neural-MLP learns a good model very quickly, reaching close to its best residual in as little as 5000 training iterations; this is similar to what we observed in the Robust Kalman Filtering models. In contrast, the model learnt by SCS+Neural-MLP for fixed source matrix distributions learns slowly and gradually over (at least) 50000 training iterations. Taken together, these results suggest that linear dynamical systems may be a class of optimization problems that benefit from neural acceleration.

## 7 Conclusion

Our work introduces and implements a neural acceleration framework that learns to accelerate fixed-point conic optimization problems drawn from a distribution; our focus is the amortized optimization setting where similar optimization problems are going to be solved repeatedly. We achieve a $10\times$ improvement in the accuracy on Kalman Filtering problem instances (drawn from a distribution). Through a series of problem and distribution modifications, we isolate a few factors that make neural acceleration more beneficial on second-order cone optimization problems (e.g., the presence of equality constraints, the amount of randomness). Our experiments suggest that linear dynamical systems may be a class of problems that can benefit from neural acceleration.

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

## A    Dynamics matrices for Kalman filtering

For ease of reference, we include here the full dynamics matrices from Diamond & Boyd (2022) used in the robust Kalman filtering problem.

$$
F = \begin{bmatrix} 1 & 0 & (1-\frac{\gamma}{2}\Delta t)\Delta t & 0 \\ 0 & 1 & 0 & (1-\frac{\gamma}{2}\Delta t)\Delta t \\ 0 & 0 & 1-\gamma\Delta t & 0 \\ 0 & 0 & 0 & 1-\gamma\Delta t \end{bmatrix} \quad G = \begin{bmatrix} \frac{1}{2}\Delta t^2 & 0 \\ 0 & \frac{1}{2}\Delta t^2 \\ \Delta t & 0 \\ 0 & \Delta t \end{bmatrix} \quad H = \begin{bmatrix} 1 & 0 & 0 & 0 \\ 0 & 1 & 0 & 0 \end{bmatrix}
$$

## B    Hyperparameter Sweep and Models Learnt

Table 3 and Table 4 show the range of parameters used in the hyperparameter sweep for SCS+Neural (with recurrent models) and SCS+Neural-MLP (with MLP models) respectively.

## C    Additional Variations

In this appendix, we include additional variations on the Lasso problems that provide insight into how the modifications affect whether an optimization problem benefits from neural acceleration.

Table 3: Parameters used for hyperparameter sweep of SCS+Neural with Recurrent Models

| Neural Model | |
|---|---|
| - use initial hidden state | True, False |
| - use initial iterate | True, False |
| Initializer: | |
| - hidden units | 128, 256, 512, 1024 |
| - activation function | relu, tanh, elu |
| - depth | $[0 \ldots 4]$ |
| Encoder (MLP applied before recurrent cell): | |
| - hidden units | 128, 256, 512, 1024 |
| - activation function | relu, tanh, elu |
| - depth | $[0 \ldots 4]$ |
| Decoder (MLP applied after recurrent cell): | |
| - hidden units | 128, 256, 512, 1024 |
| - activation function | relu, tanh, elu |
| - depth | $[0 \ldots 4]$ |
| - weight scaling | $[2.0, 128.0]$ |
| Recurrent Cell: | |
| - model | LSTM, GRU |
| - hidden units | 128, 256, 512, 1024 |
| - depth | $[1 \ldots 4]$ |

| Adam | |
|---|---|
| learning rate | $10^{-4}$, $10^{-2}$ |
| $\beta_1$ | 0.1, 0.5, 0.7, 0.9 |
| $\beta_2$ | 0.1, 0.5, 0.7, 0.9, 0.99, 0.999 |
| cosine learning rate decay | True, False |

| Misc | |
|---|---|
| max gradient for clipping | 10.0, 100.0 |
| batch size | 16, 32, 64, 128 [Lasso] |
| | 5, 10, 25, 50 [Kalman filter] |

Table 4: Parameters used for hyperparameter sweep of SCS+Neural-MLP

| Neural Model | |
|---|---|
| - use initial hidden state | True |
| - use initial iterate | True |
| - use overparametrization | True, False |
| MLP: | |
| - hidden units | 1024, 1280, 2560, 5120, 7680, 10240, 15000 |
| - activation function | relu |
| - depth | $[2 \ldots 4]$ |
| - number of overparameterization layers | $[1 \ldots 4]$ |
| Access sets: | |
| | - Various subsets of $[1, 2, 5, 8, 10, 15, 20, 30, 40]$ |
| | - Every $x$ iterations, for x in $[3, 5, 10]$ |
| Residual Intervals: | |
| | - $[3, 5, 10, 15, 20, 30]$ |
| | - Every $x$ times the gap between consecutive elements in the access set, for $x = [1, 1.5, 2, 3]$ |
| | - A starting value $x$ for $r_0$, where $x \in [5, 10, 15, 20]$, then $2x$ after $y_1$ accesses, $y_1 \in [1 \ldots 4]$, optionally $4x$ after $y_2$ accesses, $y_2 \in [1 \ldots 4]$ |

| Adam | |
|---|---|
| learning rate | $[10^{-4}, 10^{-2}]$ |
| $\beta_1$ | 0.9 |
| $\beta_2$ | 0.999 |
| cosine learning rate decay | False |

| Misc | |
|---|---|
| max gradient for clipping | 100.0 |
| batch size | 32, 64, 128 [Lasso] |
| | 25, 50 [Kalman filter] |

## C.1   Changing Solution Sparsity in Original Distribution

In this experiment, we show that the trends described in Section 6.1 essentially hold when the sparsity of the solution changes. Recall that, in our distribution above, only 10% of the variables (1 variable with $p = 10$) are set to non-zero values in $z^*$. Here we show the effect of reducing the sparsity of the solution. We define the fraction of non-zero variables in the solution $z^*$ as the *density* of the solution. In the experiments that follow, we see how the previous trends change as the density $\rho$ increases from 0.1 to 0.3 and 0.5.

Our experiment shows that the results from Section 6.1 change only slightly when the density of $z^*$ is increased to 0.3 and 0.5. Figure 10 shows the results of learning on these distributions for SCS, SCS-AA and SCS+Neural-MLP. We note that the increased density allows for slightly increased performance of SCS+Neural-MLP, i.e., at $\rho = 0.5$, the residuals of SCS-Neural now reach 5e-3 before they no longer improve over SCS.

## C.2   Fixed Solution/Noise Matrix

In this section, we include additional experiments from Section 6.2, showing that keeping the noise $w$ fixed or the solution $z^*$ does not result in a distribution where neural acceleration is beneficial.

Table 5: Recurrent models learnt for Lasso and Robust Kalman Filtering

| Lasso | |
| --- | --- |
| Adam | |
| learning rate | 0.000177 |
| $\beta_1$ | 0.1 |
| $\beta_2$ | 0.9 |
| cosine learning rate decay | False |
| Misc | |
| max gradient for clipping | 16.9160 |
| batch size | 64 |
| Neural Model | |
| - use initial hidden state | True |
| - use initial iterate | True |
| Initializer: | |
| - hidden units | 512 |
| - activation function | relu |
| - depth | 2 |
| Encoder (MLP applied before recurrent cell): | |
| - hidden units | 512 |
| - activation function | relu |
| - depth | 0 |
| Decoder (MLP applied after recurrent cell): | |
| - hidden units | 1024 |
| - activation function | tanh |
| - depth | 4 |
| - weight scaling | 10.5596 |
| Recurrent Cell: | |
| - model | LSTM |
| - hidden units | 512 |
| - depth | 2 |

| Robust Kalman filtering | |
| --- | --- |
| Adam | |
| learning rate | 0.000337 |
| $\beta_1$ | 0.1 |
| $\beta_2$ | 0.1 |
| cosine learning rate decay | False |
| Misc | |
| max gradient for clipping | 11.7536 |
| batch size | 25 |
| Neural Model | |
| - use initial hidden state | True |
| - use initial iterate | True |
| Initializer: | |
| - hidden units | 512 |
| - activation function | tanh |
| - depth | 2 |
| Encoder (MLP applied before recurrent cell): | |
| - hidden units | 256 |
| - activation function | relu |
| - depth | 3 |
| Decoder (MLP applied after recurrent cell): | |
| - hidden units | 1024 |
| - activation function | relu |
| - depth | 3 |
| - weight scaling | 12.8132 |
| Recurrent Cell: | |
| - model | GRU |
| - hidden units | 256 |
| - depth | 1 |

Table 6: MLP models learnt for Lasso and Robust Kalman filtering

| Lasso | |
| --- | --- |
| Adam | |
| learning rate | $10^{-4}$ |
| $\beta_1$ | 0.9 |
| $\beta_2$ | 0.999 |
| cosine learning rate decay | False |
| Misc | |
| max gradient for clipping | 100.0 |
| batch size | 64 |
| Neural (MLP) Model | |
| - use initial hidden state | True |
| - use initial iterate | True |
| - use overparametrization | False |
| Initializer MLP: | |
| - hidden units | 2560 |
| - activation function | relu |
| - depth | 4 |
| - number of overparameterization layers | 0 |
| Acceleration MLP: | |
| - hidden units | 5120 |
| - activation function | relu |
| - depth | 4 |
| - number of overparameterization layers | 0 |
| Access set: | [1, 2, 5, 10, 15] |
| Residual Intervals: | $\{r_0{=}5$ |
| | $r_1, r_2, r_5 = 10,$ |
| | $r_{10}, r_{15}{=}20\}$ |

| Robust Kalman Filtering | |
| --- | --- |
| Adam | |
| learning rate | $10^{-4}$ |
| $\beta_1$ | 0.9 |
| $\beta_2$ | 0.999 |
| cosine learning rate decay | False |
| Misc | |
| max gradient for clipping | 100.0 |
| batch size | 50 |
| Neural (MLP) Model | |
| - use initial hidden state | True |
| - use initial iterate | True |
| - use overparametrization | True |
| Initializer MLP: | |
| - hidden units | 5120 |
| - activation function | relu |
| - depth | 2 |
| - number of overparameterization layers | 1 per hidden/output layer |
| Acceleration MLP: | |
| - hidden units | 5120 |
| - activation function | relu |
| - depth | 2 |
| - number of overparameterization layers | 1 per hidden/output layer |
| Access set: | [1, 2, 5, 10, 20] |
| Residual Intervals: | $\{r_0, r_1, r_2 = 15,$ |
| | $r_5, r_{10}, r_{20}{=}30\}$ |

### C.2.1 Fixed Noise Matrix

Our next experiment examines how well the model learns when the noise is kept fixed. In this experiment, we generate one fixed $w$, and then use it to generate all problem instances similar to the original distribution, i.e., we draw $F$, $z^*$ from the same distributions as the original, and then use our fixed $w$ to generate $g = Fz^* + w$. $\mu$ is generated as before.

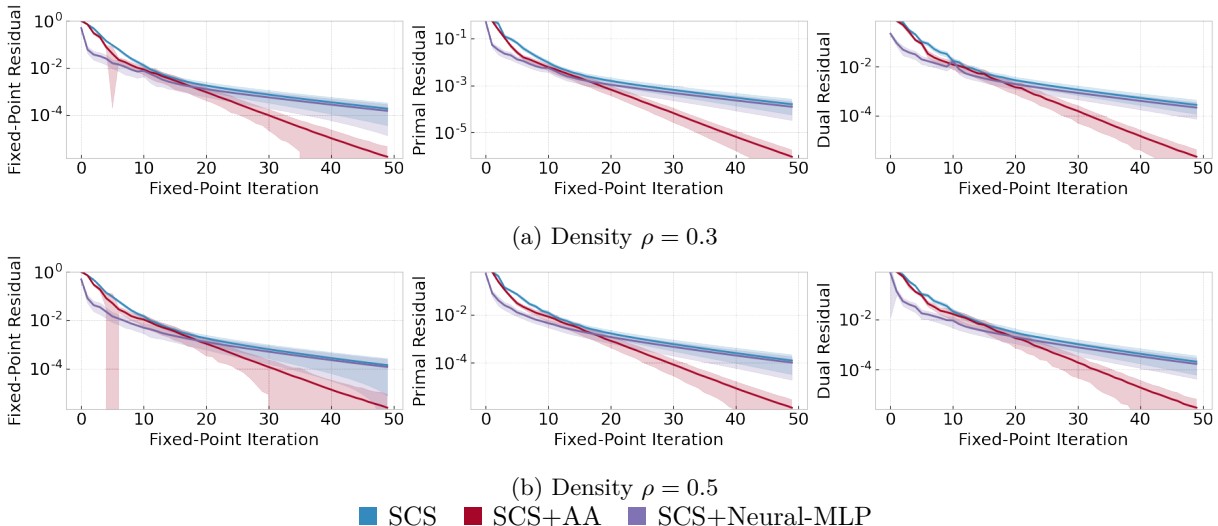

(a) Density $\rho = 0.3$

(b) Density $\rho = 0.5$

■ SCS  ■ SCS+AA  ■ SCS+Neural-MLP

Figure 10: Original Lasso distribution with increased density $\rho$ (Appendix C.1). Learned models use $A = \{1, 2, 5, 10\}, r_i = 10$ for all $i$.

Figure 11a shows the results for SCS, SCS-AA and SCS+Neural-MLP for this distribution. We note that the improvement of SCS+Neural-MLP over SCS here is again modest, only to a fixed-point residuals around $10^{-3}$; indeed, the improvement is a little less than those in Figure 5. This is consistent with our earlier finding – a problem distribution where the noise is fixed is no better than a problem distribution where the noise is much smaller than the required residuals.

### C.2.2 Fixed Solution

Our next experiment examines how well the model learns when the original solution $z^*$ is kept fixed. In this experiment, we generate one fixed $z^*$, and use it to generate all the problem instances as we did in our original distribution, i.e., we draw $F, w$ from the same distribution as we did originally, and then use our fixed $z^*$ to generate $g = Fz^* + w$. Again, $\mu$ is generated as before. Again, we generate datasets with 10 different solutions, and learn models on each of them.

Figure 11b shows the results for SCS, SCS-AA, SCS+Neural-MLP aggregated over the 10 different datasets. We see that again, SCS and SCS-AA are similar to the previous distributions, but SCS-Neural is now no longer able to learn much. Indeed, the performance of SCS-Neural is now similar to that in the original Lasso distribution (Figure 3), where the most substantial improvement over SCS comes at fixed-point iteration 5 with a residual of $10^{-2}$.

## D  Importance of $\tau$ Normalization

In this appendix, we show the importance of normalizing the fixed-point residual norms for the loss by the $\tau$ conditioning factor in SCS. Figure 12 shows the residuals obtained for Lasso when SCS+Neural does not use $\tau$ normalization in the objective. The primal/dual residuals are significantly worse than SCS and SCS+AA. The fixed-point residual shows an initial improvement, but finishes worse. As discussed in Section 4, this happens when SCS+Neural achieves a low loss by simply learning a low $\tau$.

We verify this empirically examining how $\tau$ changes over the fixed-point iterations. Figure 13 shows the mean and standard deviation of the learned $\tau$ values, averaged across all test instances and across runs with all seeds. Note that SCS and SCS+AA quickly find their $\tau$ (by iteration 3-4), and deviate very little from it. SCS+Neural, however, starts at a very low $\tau$ that slowly increases; this results in very low initial fixed-point residuals (and thus a better loss for $g_\theta^{\text{acc}}$), but poor quality solutions with high primal/dual residuals.

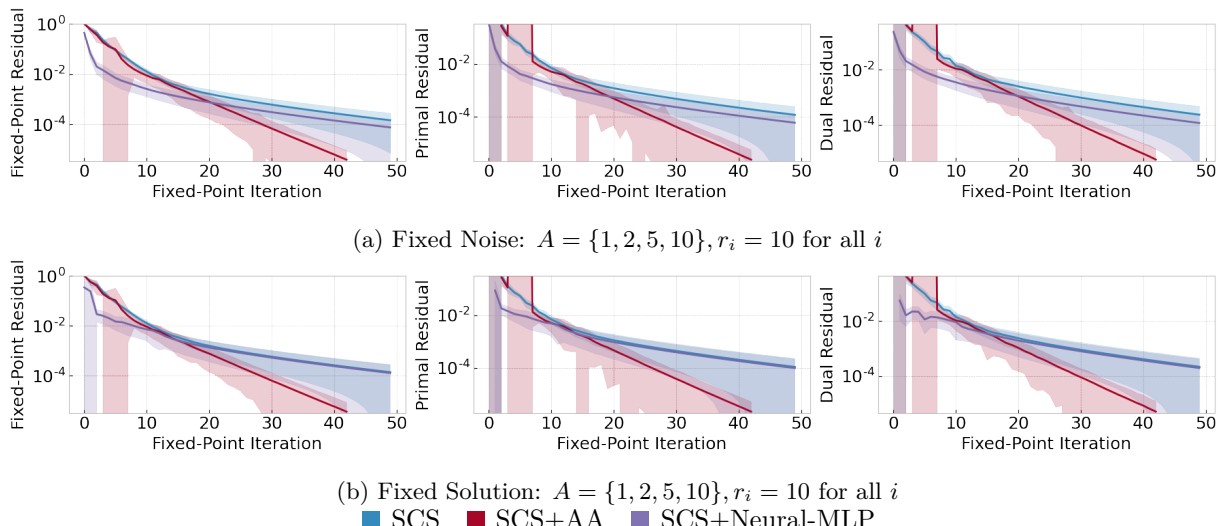

(a) Fixed Noise: $A = \{1, 2, 5, 10\}, r_i = 10$ for all $i$

(b) Fixed Solution: $A = \{1, 2, 5, 10\}, r_i = 10$ for all $i$

■ SCS  ■ SCS+AA  ■ SCS+Neural-MLP

Figure 11: Reducing the randomness in the Lasso problem distribution by keeping the noise or the solution fixed (Appendix C.2)

.

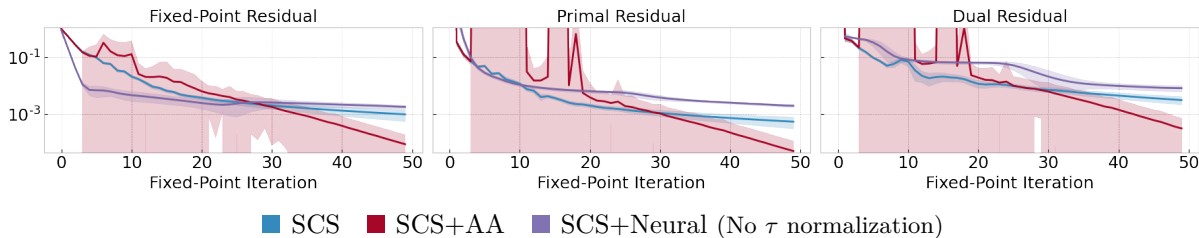

■ SCS  ■ SCS+AA  ■ SCS+Neural (No $\tau$ normalization)

Figure 12: Lasso without $\tau$ normalization: a failure mode of neural acceleration (that SCS+Neural overcomes with design), see Appendix D.

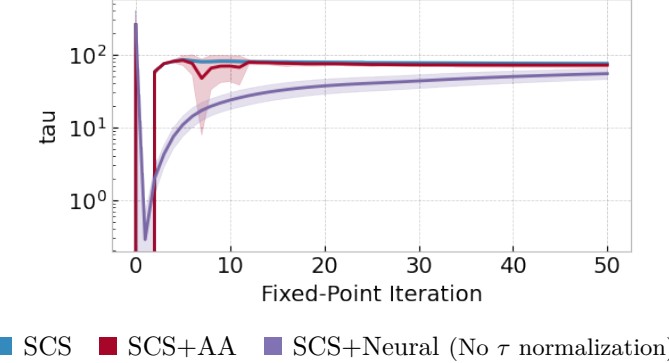

■ SCS  ■ SCS+AA  ■ SCS+Neural (No $\tau$ normalization)

Figure 13: We observe that without $\tau$ normalization, a failure mode of neural acceleration is that it learns to produce low $\tau$ values that artificially reduce the fixed-point residuals and does not solve the optimization problem well (Appendix D).

