# OpenReview forum: "Neural Fixed-Point Acceleration for Second-order Cone Optimization Problems"
_TMLR — Rejected by TMLR_

### Review · Reviewer_Bd8t · 2022-08-26

**Summary Of Contributions:**

This paper applies the "learning to learn" framework to continuous fixed-point problems to accelerate the convergence of the solvers. Specifically, the authors consider variants of Lasso and Kalman filtering problems and use Splitting Conic Solver (SCS) for these problems. By doing experiments on synthetic data, the authors showed that their neural accelerator trained using the "learning to learn" framework sometimes outperforms traditional SCS and its variants in convergence speed or accuracy. They also did ablation studies and found some important factors for their neural accelerator to perform better than traditional methods. These factors include limited model access, less random distributions of coefficient vectors/matrices, and more restrictive cone structures. Therefore, the authors concluded that linear dynamical systems might benefit more from this learning-to-learn framework.

**Broader Impact Concerns:**

This paper focuses on a general problem in optimization, and I do not see any immediate concerns about the ethical implications of the work.

**Requested Changes:**

- It would be better if the authors could provide more high-level insights and discussions about their results that could generalize to other settings. This is critical and can improve the significance of this paper.

- It might be better if the authors could explain more about the motivations/justifications behind their experimental design choices.

- Could the authors provide the runtime for the fixed-point iterations using different methods? If the neural method is much slower in runtime, the acceleration might be less useful in practice.

[Typos/Minors]

- Section 3.3 Lasso, $w\sim\mathcal{N}(0,0.1)$ -> $w\sim\mathcal{N}(0,0.1I)$

- Section 3.3 Huber function, $||a||_2$ -> $||a||_2^2$

- Section 4.3.2 Paragraph 2 (Linear System Solve), "solve and its derivative" -> "and solve its derivative"

- Section 4.4.2, line 3, "SCS+" -> "SCS+Neural"

- Figure 1(a) missing numbers on the x-axis

- Section 5.2 Line 2, "we found more efficient" -> "we found it more efficient"

- Section 6.3.1 Paragraph 2, "higher residuals can likely because" -> "higher residuals can likely be because"

- The conclusion section is almost the same as the abstract except for the last sentence. It might be better if the authors could paraphrase one of them.

**Strengths And Weaknesses:**

Strengths:

- This paper is generally clearly-written. The sections are well-organized, and the notations are well-defined. The problem settings are described in detail, making them easy to understand.

- The authors did many experiments to study the performance of their proposed methods, and the details of these experiments are provided, making their results convincing.

- This paper provided ablation studies to analyze the influence of a few factors (model access, distribution of coefficients, cone structure) on the performance of neural acceleration.

Weaknesses:

- Most conclusions from this paper seem to hold specifically for second-order cone optimization problems, which makes the results of this paper somewhat limited.

- There might not be enough justifications for the experimental settings used in this paper. For instance, the authors are performing 50 iterations, and I wonder whether the results will be different if the number of iterations changes. Besides, for the hyperparameter sweep, it might be a bit unclear how the sweep is performed, is it a grid search?

---

> ### Author Response · Authors · 2022-09-16
> **Author Response to Reviewer Bd8t**
>
> We thank the reviewer for their review. Below, we answer their questions and address the weaknesses they highlighted. We are in the process of revising our paper to include these changes and would be grateful for any additional feedback.
>
> >  * There might not be enough justifications for the experimental settings used in this paper. For instance, the authors are performing 50 iterations, and I wonder whether the results will be different if the number of iterations changes. Besides, for the hyperparameter sweep, it might be a bit unclear how the sweep is performed, is it a grid search?
>
> Our goal in neural acceleration is to use the trained model/distribution to get a fast approximate solution, rather than a highly accurate solution. We chose 50 iterations because on our test problems, SCS was able to achieve 2-3 orders of magnitude reduction in residuals by 50 fixed-point iterations. We would expect that for some problems (e.g., Lasso), SCS+AA can provide a more accurate solution if the number of allowed fixed-point iterations is greatly increased, as by then, AA will have a much longer/better history to use, and therefore, better estimate an accelerated iterate.
>
> Hyperparameter search: in Section 4, we use a random search for the hyperparameter sweep, for which the sweep parameters are listed in the Appendix (Table 3). In Section 5, we first performed a number of ablations to narrow the hyperparameter search space, and then performed random search, for which also the sweep parameters are listed in Appendix (Table 4).
>
> > * It would be better if the authors could provide more high-level insights and discussions about their results that could generalize to other settings. This is critical and can improve the significance of this paper.
>
> Our neural acceleration framework is implemented for SCS, which can solve any conic optimization problem (and thus, any convex optimization problem). However, our experiments are focused on second-order cone optimization problems, and primarily suggest that linear dynamical systems may be a class of problems that benefit from neural acceleration. Linear dynamical systems have many applications in circuit analysis, signal processing, robotics, control systems, etc.
>
> We believe that a similar improvement may be achieved in other second-order cone optimization problems with equality constraints (which produce the zero cones), and there are many engineering applications of second-order cone optimization problems, see e.g., the following references [[1]](https://web.stanford.edu/~boyd/papers/pdf/socp.pdf)  and [[2]](http://www.stat.uchicago.edu/~lekheng/courses/310w13/socp.pdf).  We hope that our results will also motivate research into more complex optimization problems with similar characteristics.
>
> > * It might be better if the authors could explain more about the motivations/justifications behind their experimental design choices
>
> Above, we have described the motivation for the choice of 50 iterations, and how we conducted the hyperparameter search; we will add these to the paper. If the reviewer wishes us to add any other specific detail about experimental design choices, please let us know and we would be happy to do so.
>
> >  * Could the authors provide the runtime for the fixed-point iterations using different methods? If the neural method is much slower in runtime, the acceleration might be less useful in practice.
>
> Our current wall-clock run-times do not compete with SCS, since we implemented an unoptimized version in Python for easier exploratory analysis – our focus was to demonstrate a proof-of-concept by learning models and identifying problem classes that may benefit from neural acceleration. In contrast, SCS has an extremely optimized implementation  in C. Our wall-clock run-time for Kalman Filtering, for example, is around 4 seconds for a batch of 64 with 50 iterations – while each model access takes only 10ms, the fixed-point iterations by themselves take a total of 1-1.5s. By contrast, on our current problem sizes, SCS takes 1-2ms on average, and SCS+AA takes 3ms on average. Likewise, for Lasso, our implementation takes 2.3s per batch of 50 for 50 iterations, while SCS and SCS+AA take 3ms and 4ms on average.
>
> While in this case the neural network evaluation takes longer than the SCS solution, this changes when the problem sizes become larger. For very large problem sizes, neural network evaluations can execute faster than the corresponding matrix inversions, especially for large ones inverted as part of the AA procedure in SCS. We believe an eventual wall-clock runtime improvement is possible with neural acceleration on the appropriate class of problems, and view our work (with a significant improvement in the number of iterations) as only a first step towards it.

---

> > ### Comment · Reviewer_Bd8t · 2022-09-17
> > **Update after author response**
> >
> > Thank the authors for the detailed response! I have read all the other views and the authors' responses to these. The response partially addressed my concerns, especially regarding the design choices and details of the experiments in this paper. However, some of my concerns still hold, and I agreed with some other weaknesses from the other reviews. I will list my main concerns at the current stage below:
> >
> > 1. The clarity of this paper needs to be improved. As mentioned by the other reviewers, many words and statements need to be further clarified for readability. For instance, it might be better to provide rigorous definitions and relationships between the problems (cone optimization problem, self-dual embedding, etc.), unambiguous descriptions of the algorithms, detailed experimental settings, and more interpretations of their results. This could require a large revision of the texts, and it would be hard to give a recommendation before the authors upload the revised manuscript.
> >
> > 2. As mentioned in the weakness part of my review, most conclusions in this paper seem to be specifically for second-order cone optimization problems. Since there is no theoretical guarantee of generalizing this acceleration ability to more settings, the significance of the results might be limited.
> >
> > 3. The wall-clock run-time of this algorithm might be too slow to be considered an acceleration. The authors claimed that their method could potentially be faster than previous methods when the problem size becomes larger, and I wonder whether there are some justifications why this neural method still performs better than previous ones on larger scale.

---

> > > ### Author Response · Authors · 2022-09-24
> > > **Revised version uploaded**
> > >
> > > We thank the reviewer for their feedback, and have uploaded a new version of paper. We have made many changes to Section 3, 4, and 5 to improve clarity, and have included a discussion on wall-clock run-time in Section 5.3.3. We hope that the reviewer finds that the new paper addresses their concerns and meets the [TMLR evaluation criteria](https://jmlr.org/tmlr/editorial-policies.html) of having well-supported claims (i.e., of being able to accelerate second-order cone optimization problems, as measured by the number of fixed-point iterations used).

---

> > > > ### Comment · Reviewer_Bd8t · 2022-09-25
> > > > **Update after Revision**
> > > >
> > > > Thank the authors for revising the paper. The clarity of this paper improved a lot in the updated version with the added definition, background, explanation, and discussions. It addresses most of my concerns about clarity.
> > > >
> > > > A minor suggestion: It might be better to provide some information about the hardware that you use to run the experiments when talking about the wall-clock runtime.

---

> > > > > ### Author Response · Authors · 2022-09-27
> > > > > **Author response to Reviewer Bd8t**
> > > > >
> > > > > Our experiments are run on Intel Xeon CPU E5-2698 v4 2.20GHz with 80 cores, with Tesla V100-SXM2-32GB GPU (using one GPU per experiment).
> > > > >
> > > > > We also discovered an error in our previous description of the model access time; we had made an error in converting seconds to milliseconds. We now also re-measured the wall-clock run-time on an unloaded server.  After fixing the error, we found that the model access time (for each inference) is between 0.6ms-0.8ms, rather than the 10ms we had mentioned earlier.  We apologize for this error. We have now fixed the error in the draft and uploaded the corrected version. The changes can be found on page 14-15 (last paragraph of page 14, and first paragraph of page 15). We thank the reviewer for their question, which led us to finding the error.
> > > > >
> > > > > To improve clarity in the draft, we have now changed how we describe the model access time. We now describe the total model access time for solving the problem instance (recall there are 5 acceleration model accesses + 1 initializer model access to solve a problem instance). Our total model access time ranges between 4ms-4.5ms (and as described earlier, these model accesses in SCS+Neural-MLP remain batched).

---

### Review · Reviewer_SmAu · 2022-08-31

**Summary Of Contributions:**

This paper proposes a framework of computing fixed-point in solving second-order cone optimization problems drawn from a distribution. The proposed framework, aiming at an acceleration in finding the fixed-point, incorporates neural networks, e.g, MLP and LSTM/GRU, as modules in the fixed-point iterations. This framework is experimentally applied to two convex cone optimization problems (i.e., linear regression with Lasso regularization, and Kalman Filtering).

**Broader Impact Concerns:**

No concerns on ethical implications

**Requested Changes:**

1. Add theoretical analysis to show convergence guarantee of the proposed framework, and show it actually find a desired fixed-point. If possible, also show that the founded fixed-point is a solution to the corresponding optimization problem.
2. Add theoretical analysis to verify that the proposed framework finds the fixed-point faster than previous methods.
3. Add details of implementation. Especially, clarify the network architecture, how does the network trained, and more details in the Algorithms.
4. The paper should also be revised so that it will be clearly written.



**Strengths And Weaknesses:**

The strength of the paper lies in the novel idea of utilizing deep learning models in generating/predicting the next fixed-point iteration. Another good point is that recurrent neural network models are used, to be aligned with the sequential property of the fixed-point iterations.

However, I found the paper is problematic in several aspects:

[Acceleration] The paper claims the proposed framework is an acceleration in finding the fixed-point. However, I cannot find any theoretical analysis (or even discussions) throughout this paper that supports this claim. By looking at the proposed algorithms and previous solvers (SCS, SCS+AA), I don’t think they are easily comparable: the proposed algorithm involves neural networks as sub-modules, while SCS directly inverts matrices. In what sense can we say the neural networks are faster, especially noting that the neural networks need an inner loop for training and usually take a long time to be well trained?
>I notice that the authors have figures (e.g., figure 1) showing that the SCS-Neural has a lower curve in the early stage. However, the plot is on a number of fixed-point iteration basis, and the time consumption or computational costs during one iteration is hidden. Also, the time of neural network training is not shown.

Hence, I don’t see any evidence indicating that the proposed framework is an acceleration.

[Convergence/reliability] Another major issue of the paper is that there is no convergence analysis of the proposed framework. Without convergence, it is not clear whether the proposed framework can actually find a fixed-point. In this sense, I don’t think it is well established that the proposed framework solves fixed-point problems.

[Clarity] I don’t think the paper is well written, and I had a hard time understanding some details. I mention a few below:

> In the definition of the fixed-point problem (Sec. 4.1), there is no dependence of $f$ on the parameters $\theta$, and I am confused how $\theta$ enters the fixed-point problem. Secondly, it reads like “the fixed-point problem is defined by a context $\phi$ …”, it seems $\phi$ is a vector, and I don’t understand how a fixed-point problem is a vector. I also don’t see how $f$ depends on the context $\phi$. Overall, I don’t think the fixed-point problem is well defined.

> Many detailed implementations are missing in the paper. For example, 1), the acceleration model $g^{acc}$ takes a sequence of past iterates $x_t$ as its hidden states. How does the network $g^{acc}$ take multiple hidden states? Also how does it take the $\phi$ as an argument? An architecture of the network is necessary. 2), Algo. 2 seems not consistent with the description below it. Specifically, in step 4 and 5 in the description is not included in Algo. 2. 3), in Sec 4.3.2, how is the normalization of loss performed? (i.e., what is the value of \tau$?).

> The paper should clarify the relation among the following three: 1), convex cone problem, Eq. 1; 2), the self-dual embedding Eq.3; 3) the fixed-point problem. Specifically, Eq.1 is an optimization problem, while Eq.2 seems not an optimization problem (without minimization or maximization). How can embedding convert an optimization into a non-optimization problem? In addition, why is the fixed-point an optimal solution? Although some can be found in literature with a bit of effort, it is important to make the context self contained.

I also have other concerns/questions.

The proposed framework is only experimentally tested in two special convex cone optimization problems. Given that there is no theoretical analysis in this paper, I am concerned about the validity of the proposed framework.

With intermittent model access in Sec 5, why is the fixed-point residual still decreasing after the last model access?

In figure 1 & 2, why are the curves for the proposed methods much smoother than the others?

---

> ### Author Response · Authors · 2022-09-16
> **Author Response to Reviewer SmAu**
>
> We thank the reviewer for their review. Below, we answer their questions and address the weaknesses they highlighted. We are in the process of revising our paper to include these changes and would be grateful for any additional feedback.
>
> > [Acceleration] The paper claims the proposed framework is an acceleration in finding the fixed-point. However, I cannot find any theoretical analysis (or even discussions) throughout this paper that supports this claim. By looking at the proposed algorithms and previous solvers (SCS, SCS+AA), I don’t think they are easily comparable: the proposed algorithm involves neural networks as sub-modules, while SCS directly inverts matrices. In what sense can we say the neural networks are faster, especially noting that the neural networks need an inner loop for training and usually take a long time to be well trained?
>
> > I notice that the authors have figures (e.g., figure 1) showing that the SCS-Neural has a lower curve in the early stage. However, the plot is on a number of fixed-point iteration basis, and the time consumption or computational costs during one iteration is hidden. Also, the time of neural network training is not shown.
>
> We wish to first clarify a few details – we want to note that at inference time, along with using the trained neural network, we also apply SCS. This is captured in Algorithms 1 and 2 as the fixed-point mapping $f$. Algorithms 1 and 2 show the inference phase where we are only using the trained neural network for acceleration together with SCS; they do not show the training phase of the neural network.
>
> We also wish to clarify that we are considering the amortized optimization setting where similar optimization problems are going to be solved repeatedly. In this setting, we aim to learn from solving historical instances in the distribution (available earlier), to
> to accelerate future instances (available at test time only), rather than being agnostic to this additional structure and restarting from scratch each time. Because of this, we focus on acceleration in terms of inference time, and do not explicitly take into account training time (the cost of the training is amortized by the test instances to which it is applied, where we can obtain faster approximate solutions).  For additional details on amortized optimization, please see https://arxiv.org/pdf/2202.00665.pdf
>
> The framework is therefore an acceleration in the following sense: the trained neural network  predicts solutions that reduce the fixed-point residual better than SCS in each iteration; in this regard, it performs the same role as Anderson Acceleration. Our current results show that the trained neural network provides an acceleration in terms of the number of iterations needed to achieve a comparable solution. Each iteration in our framework includes only a model access (perhaps intermittently) in addition to the regular fixed-point iteration — here, an improvement in the number of iterations is a necessary first step to improving the wall-clock run-time.
>
> Our current wall-clock run-times do not compete with SCS, since we implemented an unoptimized version in Python for easier exploratory analysis. Our focus was to demonstrate a proof-of-concept by learning models and identifying problem/distribution classes that benefit from neural acceleration. In contrast, SCS has an extremely optimized implementation written in C. Our wall-clock (inference) run-time for Kalman Filtering, for example, is around 4 seconds for a batch of 64 with 50 iterations – while each model access takes only 10ms, the fixed-point iterations by themselves take a total of 1-1.5s. By contrast, on our current problem sizes, SCS takes 1-2ms on average, and SCS+AA takes 3ms on average. Likewise, for Lasso, our (inference) run-time is 2.3s per batch of 50 for 50 iterations, while SCS and SCS+AA take 3ms and 4ms on average. Our model training takes 1-3 days.
>
> While in this case neural network evaluation (at inference time) takes longer than the SCS solution, this changes when the problem sizes become larger, i.e., for very large problem sizes, neural network evaluations can execute faster than the corresponding matrix inversions, especially for large ones inverted as part of the AA procedure in SCS. We believe an eventual wall-clock runtime improvement is possible with neural acceleration on the appropriate class of problems, and view our work (with a significant improvement in the number of iterations) as only a first step towards it.
>
> > Hence, I don’t see any evidence indicating that the proposed framework is an acceleration.
>
> The framework provides an acceleration in terms of the number of fixed-point iterations (measured as the residuals achieved after a specified number of iterations, or equivalently, the number of iterations required to achieve a desired residual). We believe this is a necessary first step on the path to a practical use case.
>
> (Continued in the next comment due to space limits)

---

> ### Author Response · Authors · 2022-09-16
> **Author Response to Reviewer SmAu (part 2)**
>
> > [Convergence/reliability] Another major issue of the paper is that there is no convergence analysis of the proposed framework. Without convergence, it is not clear whether the proposed framework can actually find a fixed-point. In this sense, I don’t think it is well established that the proposed framework solves fixed-point problems.
>
> We have started with SCS (an established method whose convergence is proven), and we optimize the parameters of an acceleration model for a loss that makes SCS converge faster for problem instances drawn from this distribution.
>
> We acknowledge that out-of-distribution problem instances may present additional challenges. In practice, for out-of-distribution problems, we can apply mitigations. For example, we know that the fixed-point residual must always reduce; if it so happens that our model results in increasing the fixed-point residual, we can throw it away, and just run SCS. For the intermittent access model, we have another advantage — once the model accesses are completed, we only apply SCS operations. Those operations are supposed to always keep reducing the residuals since we are in a convex setting. If we are in a situation such that the residuals of SCS change very slowly for a number of iterations, we can likely detect this as well, and restart SCS from the origin.
>
> > [Clarity]
> > In the definition of the fixed-point problem (Sec. 4.1), there is no dependence of $f$ on the parameters $\theta$, and I am confused how  $\theta$ enters the fixed-point problem. Secondly, it reads like “the fixed-point problem is defined by a context  $\phi$…”, it seems $\phi$ is a vector, and I don’t understand how a fixed-point problem is a vector. I also don’t see how $f$ depends on the context $\phi$. Overall, I don’t think the fixed-point problem is well defined.
>
> We apologize for the confusion here; it comes from the overloaded definitions of the word “parameter” between the optimization and machine learning communities. We consider fixed-point problems *parameterized* by $\phi$, which is similar to other parametric optimization settings, i.e., $\phi$ defines a problem instance. However, we refer to $\phi$ as a context of the fixed-point problem to distinguish them from the parameters of the acceleration model $\theta$ ($\theta$ is not related to the parametric fixed-point computation). Further, note that $\phi$ is used as a vector only for the purposes of input to the neural network.
>
> >Many detailed implementations are missing in the paper. For example, 1), the acceleration model $g^{acc}$ takes a sequence of past iterates $x_t$ as its hidden states. How does the network $g^{acc}$ take multiple hidden states? Also how does it take the $\phi$ as an argument? An architecture of the network is necessary. 2), Algo. 2 seems not consistent with the description below it. Specifically, in step 4 and 5 in the description is not included in Algo. 2. 3), in Sec 4.3.2, how is the normalization of loss performed? (i.e., what is the value of $\tau$?).
>
> 1) $g^{acc}$ takes in a concatenation of past inputs $x_t$ and the context $\phi$. Note that in Algorithm 2, we use MLPs, so there is no “hidden state”; instead, the past inputs and the context perform the same role as the hidden state of Algorithm 1.
> 2) The description below Algorithm 2 refers to the training phase of the neural network. The framework of Algorithm 2 only shows the inference phase; we do not include the training phase there. Steps 4 and 5 are only relevant to the training phase.
> On architecture: We have included the architecture of the best models in the results of Section 5 (i.e., Section 5.3.2), as well as the hyperparameters used in the sweep in Table 4 (Appendix). For the initial results in Section 4.3.3, we have included the hyperparameter sweep in Table 3 (Appendix), but we can add the specifics of the exact models we used as well.
> 3) $\tau$ is derived from SCS, i.e., SCS sets $\tau$ when it is applied, and we use that to normalize the loss.

---

> ### Author Response · Authors · 2022-09-16
> **Author Response to Reviewer SmAu (part 3)**
>
> > The paper should clarify the relation among the following three: 1), convex cone problem, Eq. 1; 2), the self-dual embedding Eq.3; 3) the fixed-point problem. Specifically, Eq.1 is an optimization problem, while Eq.2 seems not an optimization problem (without minimization or maximization). How can embedding convert an optimization into a non-optimization problem? In addition, why is the fixed-point an optimal solution? Although some can be found in literature with a bit of effort, it is important to make the context self contained.
>
> Pages 4 & 5 of the SCS paper (long version) describe how the cone optimization problem is converted to the homogenous self-dual embedding
> https://web.stanford.edu/~boyd/papers/pdf/scs_long.pdf
> In particular, note that the section explains any non-zero solution of the self-dual embedding is optimal for the cone optimization problem, and thus the goal is only to find a non-zero solution of the self-dual embedding. We will add high-level explanations in Section 3.2 clarifying these, along with additional pointers to the relevant Sections of the SCS paper. We did not include the full descriptions to avoid confusing the reader with the details of prior work. Section 3.4 of the SCS paper derives how the fixed-point of the SCS algorithm is the optimal solution.
>
> [Additional questions]
> > The proposed framework is only experimentally tested in two special convex cone optimization problems. Given that there is no theoretical analysis in this paper, I am concerned about the validity of the proposed framework.
>
> We feel these examples are interesting and validate our claim that we can use learning to improve fixed point computation in second-order cone optimization problems. Is there a specific optimization problem that we could experiment on that would be more convincing to you?
>
> > With intermittent model access in Sec 5, why is the fixed-point residual still decreasing after the last model access?
>
> As described earlier (under [Acceleration]), the fixed-point map of SCS is also applied during neural acceleration; so after the last model access, the residuals decrease due to the application of SCS on the current iterate.
>
> > In figure 1 & 2, why are the curves for the proposed methods much smoother than the others?
>
> This is because Anderson Acceleration has well-known stability problems, especially in the early stages of optimization, see e.g.,https://web.stanford.edu/~boyd/papers/pdf/scs_2.0_v_global.pdf  We will add a sentence explaining this in Sections 4 and 5.
>
> Tangentially, the instability of Anderson Acceleration in some cases leads to another application of learned acceleration that we have thought about pursuing as a future direction. The same machinery we have here (i.e. parameterizing the update rule and finding the best parameters that make the fixed point residuals low) can be used to help improve the stability. For example, we can parameterize a module that mixes the standard fixed point update and the standard AA update in the most stable way over a class of problem instances.
>
> **Requested Changes:**
>
> For items 1 & 2: We do not have theoretical analysis for our work, similar to much of the research in learning-to-optimize literature. Instead, we have described in the earlier answers why this is a principled framework that would find the fixed-point faster, i.e. we started with SCS (an established method whose convergence is proven), and we optimize the parameters of an acceleration model for a loss that makes SCS converge faster (for problem instances drawn from this distribution.) We have also described practical mitigations to reduce the impact on out-of-distribution problems. Further, Section 6 shows our reasoning through experimentation for concluding that linear dynamical systems may benefit from neural acceleration.
>
> For items 3 & 4: we have addressed each question the reviewer has asked separately above.

---

### Review · Reviewer_muHt · 2022-09-07

**Summary Of Contributions:**

This paper develops a neural acceleration framework that learns an acceleration model to enable a faster solver for the fixed-point problem. Furthermore, the developed framework has been applied to solve the second-order cone optimization problems, which is demonstrated to achieve a 10x  performance improvement in accuracy on the Kalman filtering distribution.

**Requested Changes:**

Please refer to the weakness part. The authors may need to revise the Section 3 to give more detailed description of the problem studied in this paper.

**Strengths And Weaknesses:**

Strength:

This paper proposed a new framework for accelerating solving the fixed-point problems.
Experimental results demonstrate the faster convergence and smaller deviation of the proposed method.
The authors also provide experiments to reason about the key factors that make neural acceleration effective.

Weakness:

1. Some important descriptions are missing and the reader may be difficult to catch the point of this paper. For example, the authors need to clarify what's the definition of the fixed-point problem and why the convex cone programming can be applied to Lasso and Kalman Filtering problems.

2. Besides, the authors keep using the term "Lasso distribution" and "Kalman filtering distribution", what do these "distributions" account for?

3. Although the authors mentioned some learning-to-optimize works in the related work, I am still curious about whether the existing learning-to-optimize methods can be applied to the fixed-point problems and also give acceleration.

---

> ### Author Response · Authors · 2022-09-16
> **Author Response to Reviewer muHt**
>
> We thank the reviewer for their review. Below, we answer their questions and address the weaknesses they highlighted. We are in the process of revising our paper to include these changes and would be grateful for any additional feedback.
>
> > 1. Some important descriptions are missing and the reader may be difficult to catch the point of this paper. For example, the authors need to clarify what's the definition of the fixed-point problem and why the convex cone programming can be applied to Lasso and Kalman Filtering problems.
>
> Our work introduces and implements a neural acceleration framework that learns to accelerate fixed-point conic optimization problems drawn from a distribution; our focus is the amortized optimization setting where similar optimization problems are going to be solved repeatedly. We achieve a 10x improvement in the accuracy on Kalman Filtering problem instances (drawn from a distribution). Through a series of problem and distribution modifications, we isolate a few factors that make neural acceleration beneficial on second-order cone optimization problems (e.g., the presence of equality constraints, the amount of randomness); our experiments suggest that linear dynamical systems may be a class of problems that can benefit from neural acceleration.
>
> We introduced the definition of a fixed-point problem at the start of the Introduction. However, for additional clarity and easy reference, we will add it also to Section 3, and provide an example. Lasso and Robust Kalman filtering (as defined) can be solved by convex cone programming because they can be formulated as second-order cone programs. We will mention this in Section 3.3 to make it clear. In our implementation, we use CVXPY to convert these formulations into standard form (mentioned in Section 4.4.1) and show the cone structures in Table 1. We will add a forward reference to this Table in Section 3.3 to highlight the cone structure more concretely.
>
> > 2. Besides, the authors keep using the term "Lasso distribution" and "Kalman filtering distribution", what do these "distributions" account for?
>
> In Section 3.3, we have described how we generate random Lasso and Kalman filter problem instances. Here "distribution" refers to the probability distribution over these random instances. In Section 4.4.1, we have described the parameters we use to instantiate them for the results of Section 4 and 5. We will make it clearer that we mean a distribution over problem instances in Section 4. In Section 6, we have instantiated the same Lasso distribution with a different set of parameters initially, and then modified the distribution for each experiment; therefore, each experiment only refers to its respective distribution. We will make this clearer in Section 6 as well, and we would be grateful for any suggestions in this regard.
>
> > 3. Although the authors mentioned some learning-to-optimize works in the related work, I am still curious about whether the existing learning-to-optimize methods can be applied to the fixed-point problems and also give acceleration.
>
> Most learning to optimize works, such as https://arxiv.org/pdf/1606.01885.pdf, http://proceedings.mlr.press/v70/wichrowska17a.html, and https://proceedings.neurips.cc/paper/2016/hash/fb87582825f9d28a8d42c5e5e5e8b23d-Abstract.html, consider only unconstrained optimization problems and it’s not clear the best way of applying them to the convex and constrained conic optimization problems we consider here. We propose to connect these topics by viewing constrained optimization problems as a fixed-point procedure and learning to improve the convergence of, or accelerate, the fixed-point procedure.
>
> In theory, all of the learning to optimize methods for unconstrained optimization could be applied to any fixed point procedure by taking the perspective that we can find a fixed point ($x^\star$ s.t. $f(x^\star)=x^\star$) by solving an unconstrained optimization problem over its residuals ($x^\star=\argmin_x ||f(x)-x||_2^2$). We propose a simple preliminary way of improving fixed point computations by interleaving an MLP with the fixed point iterations (e.g. SCS updates) and hope that this perspective opens the possibility of exploring the other methods being investigated by the learning to optimize community, such as RNNs or learning symbolic updates. One other connection to existing learning to optimize works is that many iterative methods such as gradient descent can also be seen as performing fixed point iterations.

---

### Decision · Action_Editors · 2022-10-27

**Recommendation:** Reject

**Comment:**

See "Claims and Evidence" section.

**Audience:**

The topic of this paper is of high relevance to the TMLR community, and I believe many will find interest in its ideas.

**Claims And Evidence:**

The main claim of the paper is the acceleration of second-order cone optimization problems.  One of the reviewers strongly argued that this claim is not sufficiently supported, for the following reasons:

**(1)** While the authors measure acceleration by the number of iterations, the complexity of an individual iteration is not thoroughly treated, and (as the authors admit) the wall-clock run time of their approach is currently inferior compared to existing methods.

**(2)** There are no theoretical results establishing acceleration.

**(3)** On the empirical side, only two problem instances (Lasso and Kalman filtering) are evaluated.

The other reviewers generally agreed with this assessment, but felt that the "learning-to-learn" application presented in the paper is of interest to the community.  While I fully agree with the other reviewers' sentiment, I feel that this paper currently falls short of TMLR's criteria in terms of substantiation of main claims.  Each of the shortcomings (1)-(3) should on its own **not** be an impediment for publication, but when taken together, I'm afraid they lead to an unfortunately situation where the main claim of the paper is not sufficiently supported, neither empirically nor theoretically.

While point (3) is perhaps the least important, and point (2) requires an entirely new contribution, I encourage the authors to focus on point (1).  If they do not provide an implementation demonstrating wall-clock run time improvement, I think they should at least come up with convincing arguments for why the overall computational complexity (not just the number of iterations) is improved.  I am greatly looking forward to the revised version of this manuscript, and believe it will ultimately bring forth a contribution the TMLR readership will highly value.